# Texture is encoded in precise temporal spiking patterns in primate somatosensory cortex

Katie H. Long[1,2], Justin D. Lieber[3] & Sliman J. Bensmaia [1,4,5 ✉]

Humans are exquisitely sensitive to the microstructure and material properties of surfaces. In the peripheral nerves, texture information is conveyed via two mechanisms: coarse textural features are encoded in spatial patterns of activation that reflect their spatial layout, and fine features are encoded in highly repeatable, texture-specific temporal spiking patterns evoked as the skin moves across the surface. Here, we examined whether this temporal code is preserved in the responses of neurons in somatosensory cortex. We scanned a diverse set of everyday textures across the fingertip of awake macaques while recording the responses evoked in individual cortical neurons. We found that temporal spiking patterns are highly repeatable across multiple presentations of the same texture, with millisecond precision. As a result, texture identity can be reliably decoded from the temporal patterns themselves, even after information carried in the spike rates is eliminated. However, the combination of rate and timing is more informative than either code in isolation. The temporal precision of the texture response is heterogenous across cortical neurons and depends on the submodality composition of their input and on their location along the somatosensory neuraxis. Furthermore, temporal spiking patterns in cortex dilate and contract with decreases and increases in scanning speed, respectively, and this systematic relationship between speed and patterning may contribute to the observed perceptual invariance to speed. Finally, we find that the quality of a texture percept can be better predicted when these temporal patterns are taken into consideration. We conclude that high-precision spike timing complements rate-based signals to encode texture in somatosensory cortex.

---

[1] Committee on Computational Neuroscience, University of Chicago, Chicago, IL, USA. [2] Medical Scientist Training Program, University of Chicago, Chicago, IL, USA. [3] Center for Neural Science, New York University, New York, NY, USA. [4] Department of Organismal Biology and Anatomy, University of Chicago, Chicago, IL, USA. [5] Neuroscience Institute, University of Chicago, Chicago, IL, USA. ✉email: sliman@uchicago.edu

Spike timing at the level of single milliseconds has been shown to carry stimulus information in the somatosensory nerves[1]. For example, the frequency composition of skin vibrations is encoded in the phase-locked responses of individual nerve fibers[2–4], observed for frequencies up to 1000 Hz in a subpopulation of nerve fibers, namely Pacinian corpuscle-associated (PC) fibers. Temporal coding of vibratory frequency is also observed in somatosensory cortex and is particularly prominent in neurons that receive a preponderance of their input from PC fibers[5,6]. The preservation of spike timing in cortex at this level of precision is surprising given that these signals have already passed through several synapses, including the cuneate nucleus, and the thalamus[7–10]. Note, however, that similar precision is observed in rodent barrel cortex[11–14] and along the auditory neuraxis with equivalent synaptic passes[1,15–17].

By extension, the perception of texture—particularly of fine texture—is supported by a temporal code. Indeed, scanning a textured surface across the fingertip leads to the elicitation of vibrations in the skin that reflect the spatial structure of the surface[18,19] and depend on the speed at which it is scanned[20]. The frequency composition of these texture-elicited vibrations is encoded in temporally patterned responses in vibration-sensitive nerve fibers, including PC fibers[21]. As a result, afferent responses are much more informative about texture identity when spike timing is taken into consideration than when it is not[21]. The responses of neurons in somatosensory cortex, particularly those that receive strong PC input, have also been shown to exhibit temporal patterning[22]. However, the reliability of this patterning, its informativeness about texture, or its relation to perception have never been investigated. To fill these gaps, we first gauge the precision and reliability of the temporal patterning in cortical responses to texture. Second, we assess the degree to which and the temporal resolution at which texture information is encoded in temporal spiking patterns in cortex. Third, we examine how texture-specific temporal spiking patterns change with changes in scanning speed. Finally, we examine the degree to which temporal spiking patterns are predictive of the resulting texture percepts.

## Results

We recorded the responses of 141 neurons in somatosensory cortex (SC) of 2 male rhesus macaques—35 in Brodmann's area 3b, 81 in area 1, and 25 in area 2—as we scanned each of 59 diverse, everyday textures across the fingertips with precisely controlled speed and contact force[22,23] (Supplemental Table 1).

**Reliability of temporal patterns in somatosensory cortex.** A temporal spiking pattern signals the presence of a stimulus to the extent that the pattern is reliably evoked when the stimulus is presented. In the peripheral nerves, the temporal patterning of texture-evoked responses is nearly identical across multiple repeats, yielding a robust temporal code of texture identity[21]. Having observed temporal patterning in the cortical responses to textures (Fig. 1A), we quantified the reliability of this patterning across repeated presentations of the same texture and assessed its temporal fidelity. To these ends, we computed the dissimilarity between the responses of individual cortical neurons to repeated presentations of the same texture using a spike distance metric, which computes the cost of transforming one spike train into another[24]. Varying the cost of shifting spikes in time allows us to manipulate the temporal resolution of this metric: When the cost is high, even small inconsistencies in spike timing drive large dissimilarity values; when the cost is zero, distance values are driven only by differences in spike count. If responses are temporally precise across repeated presentations of the same stimulus, the pairwise dissimilarity of the responses should be low, even

when evaluated at a high temporal resolution. Even for large shifting costs, however, spike distance is driven in part by spike count. To isolate the contribution of spike timing to dissimilarity, then, we computed the spike distance metric for rate-matched simulated responses whose temporal precision could be systematically manipulated. In one case, responses were simulated using a Poisson model, thereby eliminating any information in spike timing. In the other case, a measured response was repeatedly jittered by a specific amount (Fig. 1B).

We first assessed each neuron's temporal precision by comparing the variability of its responses to that of their rate-matched jittered counterparts (Fig. 1B). For each neuron and texture, we then identified the amount of imposed jitter at which the temporal variability across simulated responses exceeded that of the measured responses (Fig. 1C–F). For each neuron, the median of the distribution of temporal resolutions across textures was then taken to be the temporal resolution (Fig. 1E). We repeated this repeatability analysis using the Poisson neurons to assess the resolution this approach would yield in the absence of temporal patterning. We found that cortical neurons produce more temporally precise responses than do rate-matched Poisson neurons (Wilcoxon signed-rank test, $Z = 9.4$, $p < 0.0001$). Furthermore, the temporal resolution (or precision) varied widely across neurons, though most neurons (75%) yielded temporal resolutions better than 5 ms (Fig. 1F).

**Rate vs timing codes.** For a temporal pattern to signal the presence of a stimulus, it must not only be reliable but also stimulus specific. With this in mind, we assessed the degree to which we could classify textures based on temporal spiking patterns. To isolate the contribution of timing to texture identification, we implemented a classifier based on pairwise correlations of single-trial time-varying responses. In brief, a texture was correctly classified to the extent that the time-varying response it evokes—the smoothed time-varying firing rate—was consistent across repeated presentations and different from responses evoked by other textures. For this analysis, we used cross-correlation as a metric of similarity between spike trains to eliminate information carried by the firing rate. By smoothing the spike trains with filters of varying width, we could vary the temporal resolution at which spiking similarity was compared across trials (Supplemental Fig. 2, Supplemental Fig. 3). In agreement with our findings from the repeatability analysis, we found that classification performance peaked at a high temporal resolution (<5 ms) in individual cortical cells (Fig. 2A) and resolutions were similar to those found in the repeatability analysis (Supplemental Fig. 4). As expected, simulated responses of Poisson neurons, which by design do not contain any temporal patterning, yielded chance classification performance (Fig. 2A, dashed lines). To further validate this approach, we verified that the classifiers' performance peaked at the resolution matching the amount of jitter introduced in the simulated responses (Supplemental Fig. 5).

Next, we compared the informativeness of rate, timing, and their combination based on classification performance. Because neurons vary in the precision of their temporal patterning, we evaluated the combination of rate and timing at each neuron's optimal resolution. We also allowed the contribution of rate and timing to vary to optimize classification performance. We found that, at the single-cell level, classification performance based on rates alone was poorer than classification based on timing alone (Wilcoxon signed-rank test, $Z = 9.7$, $p < 0.0001$) but performance with both rate and timing exceeded that with either code (Friedman's test comparing rate, timing, and their combination; $\chi^2 = 255.26$, $p < 0.0001$; post-hoc 1-sided Wilcoxon signed-rank

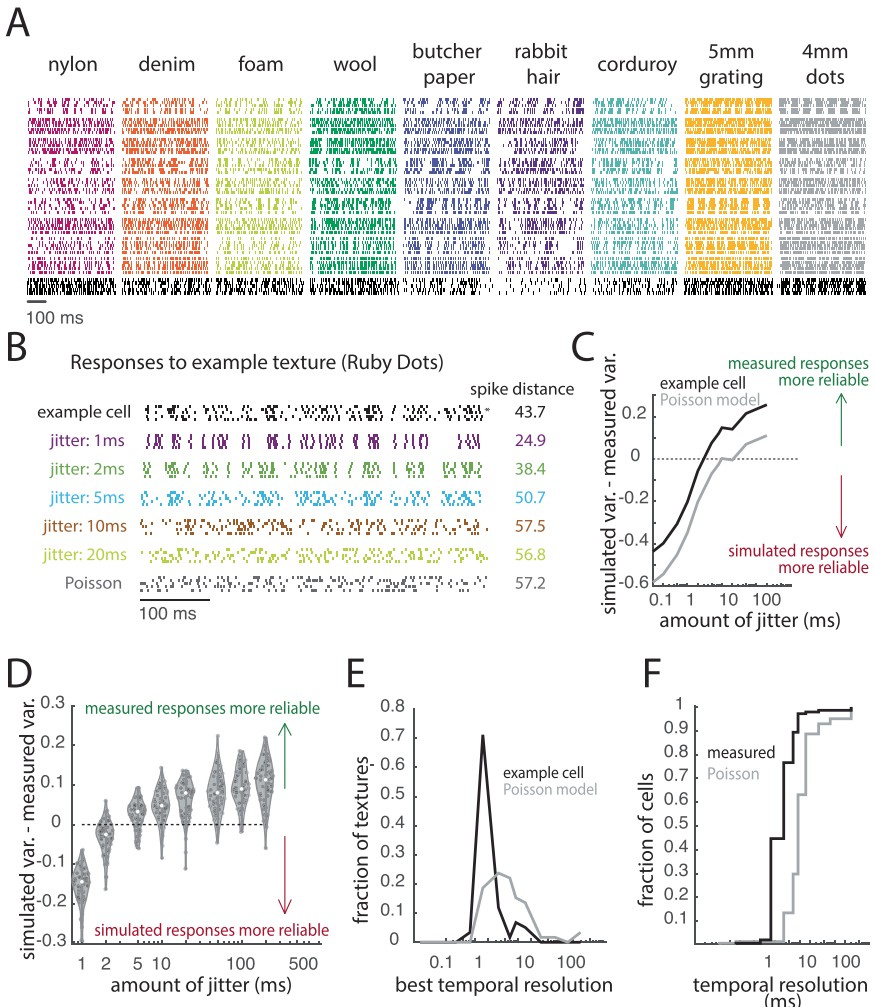

**Fig. 1 Responses in somatosensory cortex are temporally precise. A** Responses of ten example neurons to five repeated presentations of nine (of 59) textures. Each color denotes a different texture, each row denotes the response of an individual neuron across five repeated presentations of that texture. The bottom row of responses, colored in black, is from the example cell used in (**B**–**F**) of this figure. For more example responses, see Supplemental Fig. 7. **B** Response of one example neuron to 5 repeated presentations of one texture (an upholstery fabric). The asterisk indicates the trial response (trial #2) that was used to generate the simulated (jittered) responses. Colored rasters represent rate-matched simulated responses with different amounts of jitter. The gray raster is a rate-matched response from a Poisson model. Spike distances represent mean pairwise values across the five measured or simulated responses. **C** To assess the match in the variability of measured and simulated responses, we first divide the spike distance by the mean firing rate across repetitions and subtract this value from its counterpart calculated from the simulated response. The point at which this line crosses the x-intercept represents the point at which the measured responses become more temporally reliable than their simulated counterparts. The black trace is derived from the measured response of the neuron and the gray trace is derived from a rate-matched Poisson model to 'Ruby Dots'. **D** The difference in the variability of the measured and simulated responses as a function of jitter for all textures. Each point is one texture. For most textures, measured responses are more reliable than simulated responses with jitter set to 5 ms. **E** Histogram of the resolutions estimated from the responses to all repeated presentations of all 59 textures of the example neuron (black) and its rate-matched Poisson counterpart (gray). **F** Cumulative distribution of the temporal resolutions, determined using the methods shown in (**B**–**E**), of all neurons (measured, black) and their rate-matched Poisson models (Poisson, gray).

test: combined vs. rate, $z = -10.26$, $p < 0.0001$; combined vs. timing, $z = -10.26$, $p < 0.0001$; Fig. 2B), and this was true for every neuron. Note that cross-validated classification mitigates the advantage of overfitting multiple input features.

Finally, we assessed how these coding schemes scaled at the population level by assessing classification performance based on temporal and rate codes across neuronal samples of increasing size. To this end, we averaged the (inverted then standardized) correlation and rate difference for each pair of textures across the neuronal sample and then computed a weighted sum of the two distance matrices to obtain a distance metric that integrates timing and rate. We found that classifiers with both rate and timing reached higher asymptotic performance than did classifiers with rate or timing in isolation (permutation test comparing

population classification using 140 cells, $p < 0.0001$ for both combined vs. rate and combined vs. timing) and that timing-based classifiers leveled off at a lower performance level than did rate-based classifiers (permutation test, $p < 0.0001$; Fig. 2C). Classifiers with both rate and timing also reached higher performance with fewer neurons: To achieve 90% classification accuracy required 29 neurons with rate alone and 13 neurons with both rate and timing; timing alone never yielded that level of accuracy. We also evaluated the performance of a population classifier in which timing and rate were equally weighted and found performance to be nearly indistinguishable from the combination with optimized weights (permutation test comparing population classification using 140 cells, either with optimal weights or with 50/50 weights, $p = 0.69$; Supplemental Fig. 6).

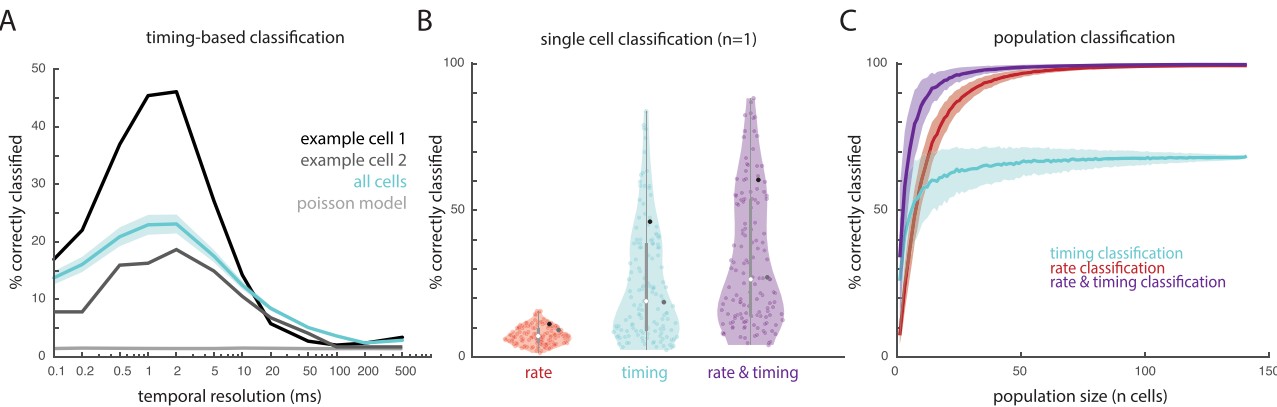

**Fig. 2 Temporal spiking patterns in somatosensory cortex carry texture information. A** Classification performance (percentage of textures correctly classified from the full texture set, comprising 59 unique textures) is best at high temporal resolutions (1–5 ms). The temporal resolution denotes the standard deviation of the Gaussian filter used to smooth the neuronal response. Performance derived from example neurons is shown in black and dark gray, mean performance across all cortical neurons is shown in blue, mean performance from rate-matched Poisson simulated neurons is shown in light gray. Simulated Poisson responses, which do not carry texture information in their timing, yield chance classification performance (1/59 textures ~ 2%). Shaded area denotes the standard error of the mean. **B** Single-cell classification performance for all 141 neurons for rate (red), timing (blue), and their optimal combination (purple). Dark points denote the example neurons shown in panel A. Violin plots show all values. Boxplots indicate median (center), interquartile range (boxes), and maximum and minimum (whiskers). **C** Mean classification performance with neuronal populations of different sizes; shaded area denotes standard deviation across 1000 iterations at each sample size. Timing-based classification (blue) yields better performance than does its rate-based counterpart (rate) for very small groups of cells, but timing-based performance levels off at a much lower level than does rate-based performance. Rate is nearly perfect with even a small population of 50–100 cells, but a combination of rate and timing (purple) is better for neuronal populations of any size and reaches 90% performance with only 13 cells (as compared to rate, which requires 29 cells).

**Temporal coding depends on submodality input.** The glabrous skin of the hand is innervated by three main classes of nerve fibers, each of which terminates in a different type of mechanoreceptor, responds to a different aspect of skin deformation[25], and makes a distinct contribution to texture coding[21,26]. PC fibers transduce texture-elicited vibrations into highly precise temporal spiking patterns; slowly-adapting afferents type-1 (SA1) fibers respond to slower, larger skin deflections and, accordingly, reflect coarse textural features in their spatial pattern of activation; rapidly-adapting (RA) fibers exhibit response and texture-coding properties that are intermediate between those of PC and SA1 fibers[21].

Individual cortical neurons receive convergent input from multiple tactile submodalities (SA1, RA, PC)[6,27,28] and exhibit highly idiosyncratic responses to texture that can be explained in part by the nature of their afferent input[22,23]. For example, cortical cells that receive dominant input from PC fibers are more likely to exhibit temporally patterned responses to textures than are cells that receive dominant input from SA1 fibers[22].

With this in mind, we investigated the relationship between submodality composition of a neuron's input and its tendency to convey texture information via precise spike timing. To this end, we first grouped cortical cells as SA1-like, RA-like, or PC-like based on their pattern of texture-evoked firing rates. In brief, we regressed the firing rates of each cortical neuron on the mean firing rates of the three classes of nerve fiber evoked by a common set of textures and identified populations of neurons with a high standardized SA1, RA, or PC regression coefficient (>0.8). Importantly, the strategy to group cortical neurons by dominant submodality did not take temporal patterning into consideration.

First, we found differences in the informativeness of spiking patterns of PC-like, RA-like, and SA1-like cortical neurons as evidenced by differences in timing-based classification performance ($n = 12$ PC-like cells, 12 RA-like cells, and 25 SA1-like cells; Kruskal–Wallis test comparing best performance of individual cells across groups, H(2) = 12.13, $p = 0.002$). Spiking patterns of PC-like cortical cells were far more informative than were those of SA1-like, as expected given the relative propensities of PC and SA1 nerve fibers to exhibit temporal patterning[3,21]

(1-sided Mann–Whitney $U$ test comparing the best classification performance of PC-like and SA1-like neurons across resolutions, $U = 330$, $p < 0.001$; Fig. 3A). RA-like cells yielded intermediate performance, but the differences did not reach statistical significance (PC-like vs. RA-like cells, $U = 171.5$, $p = 0.11$; SA1-like vs. RA-like cells, $U = 189.5$, $p = 0.02$). Notably, spike timing in PC-like, SA1-like, and RA-like cells was more informative than was that in rate-matched Poisson models (Wilcoxon signed-rank test, PC-like: $U = 78$, $p < 0.001$; SA1-like: $z = 4.4$; $p < 0.0001$, RA-like $U = 78$, $p < 0.001$). Second, the informativeness of the responses of PC-like neurons always peaked at high temporal resolutions while not all SA1-like or RA-like responses did (Fig. 3B). Note that the high temporal resolutions of many SA1-like neurons may reflect the contribution of (non-dominant) PC/RA input or the maintained temporal reliability of exceptionally precise SA1 input (Supplemental Fig. 8).

Next, we compared the temporal coding in cortex to its peripheral counterpart using a shared set of 24 textures (Supplemental Fig. 8). We found striking similarities between PC-like cortical cells and PC afferents: they yielded similar classification performance (Mann–Whitney $U$ test: $U = 126.5$, $p = 0.61$) with similar optimal temporal resolutions (median of 1 ms in both cortex and periphery). Likewise, temporal coding was weak (as indexed by timing-based classification performance) for both SA1 fibers and SA1-like cortical neurons, though spike timing in the former was more informative than in the latter ($U = 366$, $p < 0.0001$). The informativeness of timing of RA-like neurons was intermediate between that of PC-like and SA1-like neurons but far weaker than that of RA fibers ($U = 118$, $p = 0.03$).

The contrast between the three populations of nerve fibers and their downstream targets was also observed at the population level. In small populations of both PC fibers and PC-like cortical cells, timing classification exceeded rate classification (permutation test comparing rate classification to timing classification in groups of 5 cells, peripheral: $p < 0.01$, cortical: $p < 0.0001$; Fig. 3D, Supplemental Fig. 9B). In contrast, populations of SA1 fibers and SA1-like cortical cells yielded better classification performance with rate than with timing (peripheral: $p < 0.0001$, cortical:

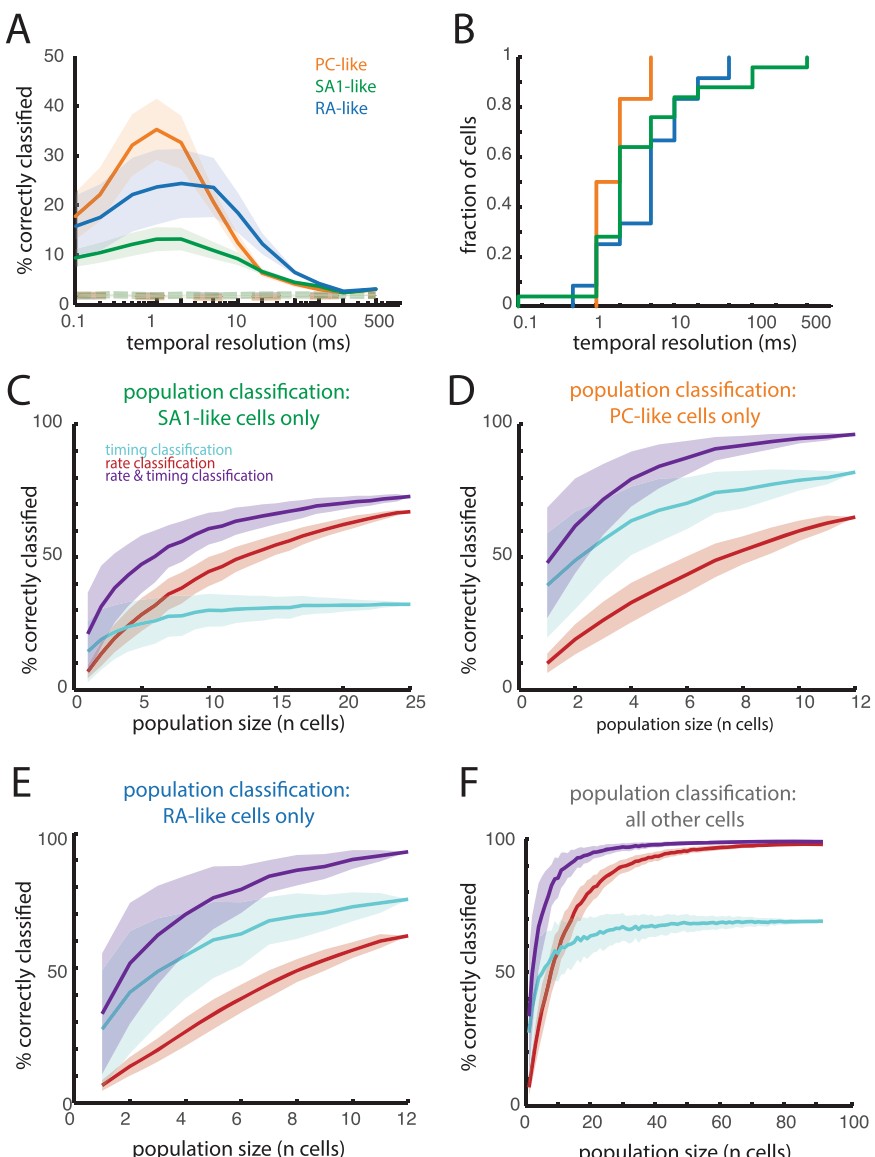

**Fig. 3 Informativeness of spike timing is related to the submodality composition of a neuron's input. A** Mean classification performance for individual PC-like neurons (orange, $n = 12$), SA1-like neurons (green, $n = 25$), and RA-like neurons (blue, $n = 12$). PC-like and RA-like responses allow for better classification than do SA1-like responses, and all are better than rate-matched simulated Poisson responses (dashed lines). Shaded regions denote the standard error of the mean across neurons. **B** Cumulative distribution of the peak temporal resolution for individual neurons. **C–F** Population classification using rate (red), timing (blue), and both (purple) for SA1-like neurons (**C**), PC-like neurons (**D**), RA-like neurons (**E**), or the remaining 92 unclassified cells (**F**). Shaded regions denote the standard deviation across 200 iterations.

$p < 0.0001$; Fig. 3C, Supplemental Fig. 9A). As was the case with their PC and PC-like counterparts, populations of RA fibers and RA-like cortical cells also yielded better classification performance with timing than with rate (peripheral: $p < 0.0001$, cortical: $p < 0.0001$; Fig. 3E, Supplemental Fig. 9C). As was found in the analysis of single cells, the informativeness of spike timing in populations of RA-like neurons was lower than was its afferent counterpart, suggesting a loss of fidelity in RA-based texture signals as they ascend the neuraxis. Neurons whose inputs were not dominated by any one modality tended to be more informative in their rates than timing (Fig. 3F).

These results suggest that the temporal coding properties of cortical neurons are to a large degree inherited from their inputs.

**Temporal precision decreases at successive stages of cortical processing.** Nerve fibers exhibit responses to vibrations that are

more precisely phase-locked than do neurons in somatosensory cortex. This loss in spike timing precision is also observed at successive stages of processing in cortex[5]. Indeed, a subpopulation of neurons in Brodmann's area 3b—the first stage of cortical processing—exhibits entrained responses to sinusoidal stimulation at frequencies up to 800 Hz; neurons in area 1, a downstream target of area 3b, are less susceptible to high-precision temporal patterning, and neurons in area 2 even less so. Accordingly, we examined whether this progressive loss of temporal precision was also observed in cortical responses to texture. As expected, the preponderance of neurons that carry information about texture in spike timing decreased at successive stages of processing, as evidenced by a decrease in timing-based texture classification across areas (Kruskal–Wallis test comparing areas 3b, 1, and 2, $H(2) = 9.37$, $p < 0.01$; post-hoc Mann–Whitney $U$ tests comparing areas 3b and 1, $U = 2.0$, $p = 0.02$, and areas 1 and 2, $U = 1.7$, $p = 0.04$; Fig. 4A, Supplemental Fig. 10). The weaker temporal patterning

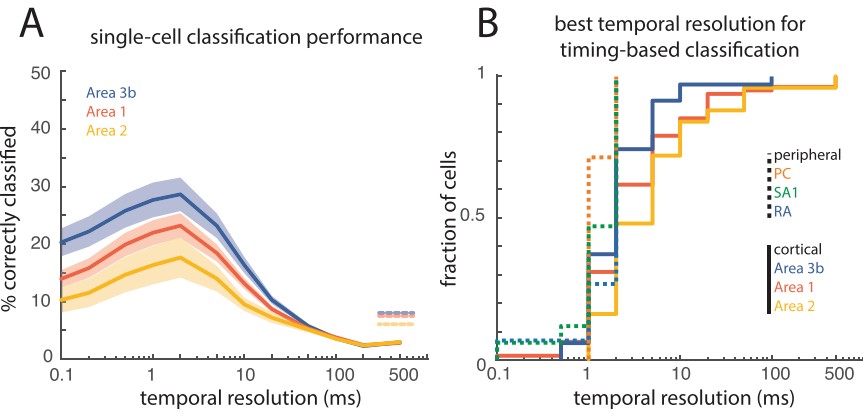

**Fig. 4 Differences in the informativeness of temporal patterns across cortical fields. A** Mean timing-based classification for individual neurons in areas 3b ($n = 35$, blue), 1 ($n = 81$, red), and 2 ($n = 25$, yellow). Shaded area represents standard error of the mean across neurons. Dashed lines on the right denote the mean classification performance based on firing rates for each area. **B** Cumulative distribution of the best decoding resolutions separated for nerve fibers (dashed lines; PC: orange, SA1: green, RA: blue) and cortical neurons (solid lines, 3b: blue, 1: red, 2: yellow).

in higher cortical areas was not accompanied by a decrease in temporal resolution, however (Kolmogorov–Smirnov test comparing temporal resolutions in areas 3b and 2, $D = 0.26$, $p = 0.23$; Fig. 4B), suggesting that, to the extent that temporal patterns propagate, their timescale is preserved. Notably, differences in the prevalence of temporally precise neurons across cortical fields are not driven by differences in submodality input. That is, the temporal patterning in area 3b is more informative than its counterpart in areas 1 and 2 despite the fact that no neurons in area 3b were classified as receiving dominant input from PC fibers and only two received dominant input from RA fibers (Supplemental Fig. 10). In other words, the incidence of PC-like neurons in areas 1 and 2 was not sufficient to overcome the overall decrement in temporal precision.

**Robustness of temporal codes across changes in scanning speed.** In the nerve, temporal spiking patterns reflect vibrations elicited in the skin as the surface is scanned across it[21]. These texture-elicited vibrations depend not only on the texture[19] but also on the speed at which it slides across the skin[20]. Indeed, texture-elicited skin vibrations systematically dilate and contract with decreases and increases in scanning speed, respectively, as do the evoked temporal spiking patterns in the nerve. In contrast, the perception of texture remains remarkably invariant to changes in scanning speed despite these speed-dependent changes in afferent responses[29–31]. The systematic effects of speed on temporal patterning can be reversed by multiplying the inter-spike intervals by the speed, thereby expressing spike trains in space rather than time[32]. To test the impact of changing scanning speed on texture-elicited temporal spiking patterns, we recorded the responses of a subset of cortical neurons as 10 textures were scanned across the fingertip at a range of behaviorally relevant speeds (60 to 120 mm/s)[33]. We then examined the degree to which the temporal patterning in the cortical response was speed-dependent and assessed whether texture information could be extracted from these temporal patterns across speeds.

First, we replicated the finding, documented for tactile nerve fibers, that texture-elicited temporal spiking patterns scale systematically with scanning speed (Fig. 5A). Indeed, spikes evoked by a given texture at different speeds could be aligned by expressing the neuronal response in space rather than time. To quantitatively assess the impact of scanning speed on the temporal code for texture, we trained timing-based classifiers on responses at one speed (training speed) and tested them at other speeds. Without warping the spike trains by speed,

classification was approximately at chance level (Fig. 5B, left; Fig. 5C). When spike trains were warped by speed, the peak performance of cross-speed classifiers was much closer to that of within-speed classifiers (Fig. 5B, right, Fig. 5C). In contrast, warping spike trains into spatial units decreased the performance of rate classifiers by overcompensating for the speed-dependent modulation of firing rates (1-sided Wilcoxon signed-rank test comparing single-cell rate classification using warped and unwarped responses, $z = 15.2$, $p < 0.0001$, Fig. 5C).

At the population level, combining rate and (warped) timing codes yielded better classification performance than did either code in isolation (permutation test comparing population classification using both rate and timing to rate alone, $p < 0.0001$, or to timing alone, $p < 0.0001$; Fig. 5D, Supplemental Fig. 12). This improvement could not be accounted for by the increase in the number of predictors (Supplemental Fig. 11), indicating that spike timing carries information beyond that carried in the spike rates, even in larger populations of neurons (Fig. 5D).

**Temporal patterning shapes texture perception.** To establish a neural code requires not only to demonstrate that the neuronal signals carry information about stimuli but also that these neuronal signals covary with the perception of these stimuli[34]. Having demonstrated that temporal patterning in cortical responses carries texture information, we thus set out to gauge whether these temporal patterns relate to the evoked textural percept. To this end, we asked human subjects to perform two psychophysical tasks: roughness scaling and dissimilarity scaling. In the roughness scaling task, each texture was presented individually and the subject provided a rating of its roughness. In the dissimilarity scaling task, a subset of thirteen textures was presented in pairs and the subject rated the dissimilarity of each pair[22] (Supplemental Table 1). While roughness ratings gauge where textures fall along a single sensory continuum, dissimilarity ratings take into account every way in which textures might differ.

First, we assessed the degree to which the perceived dissimilarity of a pair of textures could be predicted from differences in the firing rates evoked in the cortical population by the two textures. In other words, do textures feel different to the extent that they evoke different firing rates? We found that rate differences were poor predictors of dissimilarity ratings (Supplemental Fig. 13B), as might be expected as population firing rate encodes roughness[22], one of many perceptual attributes of

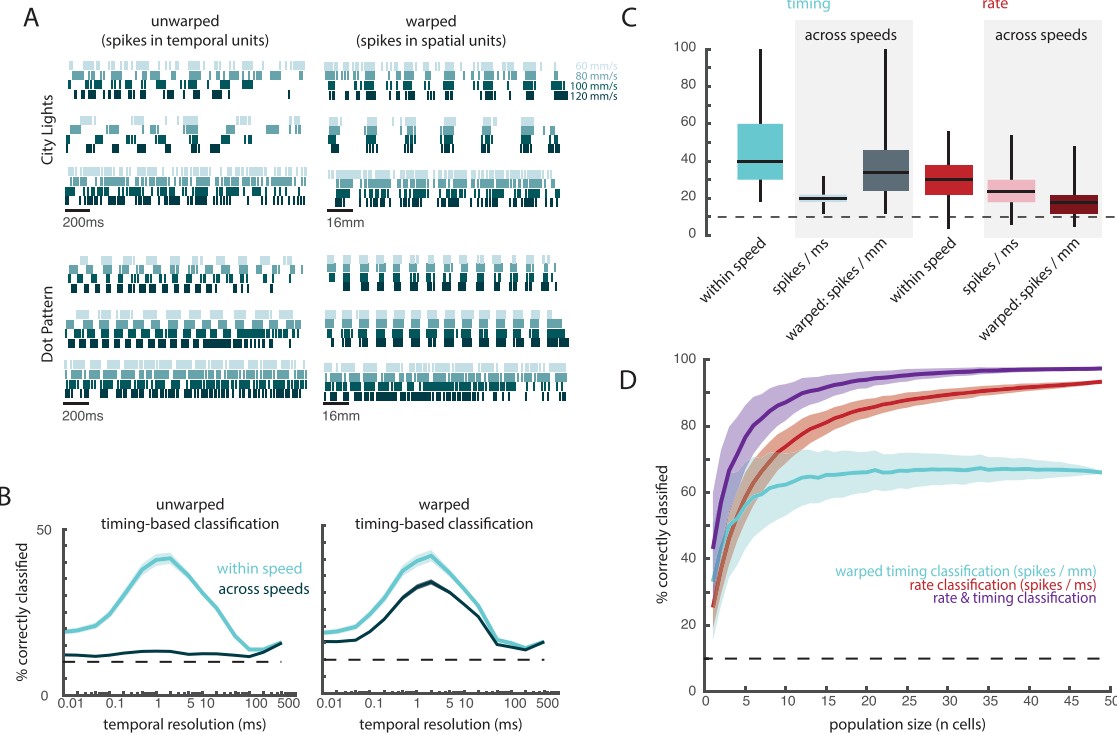

**Fig. 5 Temporal spiking patterns depend on scanning speed. A** Responses of three example neurons to two different textures (City Lights, a fabric with fine textural features and coarse ridges, as well as a dot pattern). Colors denote scanning speed, with darker colors corresponding to faster speeds. On the left, spikes are plotted across time. On the right, spikes times are "warped" such that each spike is plotted per mm of the texture rather than ms in time (by multiplying inter-spike intervals by scanning speed). **B** Timing-based classification of texture when trained on responses to textures presented at one speed (60, 80, 100, or 120 mm/s, $n = 49$ cells) and tested either within speed (cyan) or across speeds (dark blue). On the right, spike times are warped as in (**A**), and classifiers are trained and tested on these warped spike trains. Chance performance is 10% (dashed line). **C** Mean classification based on timing (blue) or rate (red), within and across speeds. Two cross-speed classifiers were assessed; light bars represent unwarped (spikes/ms), dark bars represent warped (spikes/mm) spike trains. Boxes show the median and interquartile range and the whiskers show the full range across 49 neurons and speed combinations (within speed, $n = 4$ speeds; across speeds, $n = 12$ combinations). **D** Mean cross-speed population classification based on rate (unwarped, red), timing (warped, blue), and an averaged combination of both (purple). Shaded regions denote standard deviation across 1000 iterations of randomly sampled populations of neurons.

texture[35]. We might then expect that the contribution of roughness to dissimilarity would be subsumed by the population firing rate. Consistent with this hypothesis, we found that differences in population firing rate accounted for around 20% of the variance in the dissimilarity rating, as did differences in roughness ratings (Supplemental Fig. 13B, C).

Second, we examined whether differences in temporal patterning might covary with perceptual dissimilarity. According to this hypothesis, two textures should feel dissimilar to the extent that the correlation between their PSTHs is low. Contrary to this prediction, we found this metric of temporal dissimilarity to be a poor predictor of dissimilarity ratings ($R^2 = 0.05$; Supplemental Fig. 13D). However, this analysis averaged timing differences across the entire population, disregarding the fact that neurons vary widely in their susceptibility to carry texture information in their timing. We reasoned that informative temporal patterning might be more susceptible to drive perception. To test this hypothesis, we limited the correlation-based analysis to the responses of PC-like neurons, which carry the most texture information in their temporal patterning. We found that timing differences in this neuronal population yielded far better predictions of perceived dissimilarity than did timing across the full population (1-sided Mann–Whitney $U$ test comparing the MSE of the cross-validated predictions derived from the responses of the full population and those of the PC-like subpopulation, $z = -6.8$, $p < 0.0001$), far exceeding predictions based on the firing rates of this same neuronal population

(1-sided Wilcoxon signed-rank test comparing rate and timing predictions from a subpopulation of PC-like cells, $z = -4.1$, $p < 0.0001$; Fig. 6A, Supplemental Fig. 13E, Supplemental Fig. 15). In contrast, the timing of SA1-like or RA-like neurons were poor predictors of perception (Supplemental Fig. 15).

However, because timing differences are most predictive of perception does not entail that they solely determine it. With this in mind, we investigated whether the combination of rate and timing might drive perception. To this end, we performed a multiple regression of rate and timing differences onto dissimilarity ratings, separating the regressors by cortical subpopulations (PC-like, SA1-like, and RA-like). After a systematic search through many different candidate codes (Supplemental Fig. 14), we found that the combination of timing and rate across all three cortical populations yielded better predictions than either timing or rate (Fig. 6B, C, Supplemental Fig. 14). In other words, perceived texture is shaped by both the rate and the timing of neurons across the entire cortical population.

## Discussion

**Temporal coding along the somatosensory neuraxis**. An essential question in neuroscience is whether the precise timing of spikes carries behaviorally relevant information. The somatosensory system is well suited to address this question because high-frequency deformations of the skin evoke responses in the nerve and in the brain that are reliably patterned with millisecond

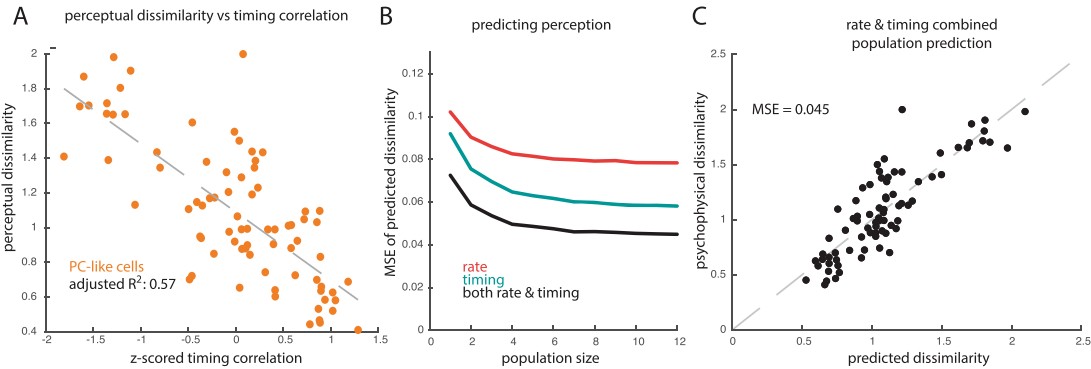

**Fig. 6 Temporal patterns predict texture perception. A** The correlation between the temporal spiking patterns evoked by texture pairs in PC-like neurons is negatively correlated with perceived dissimilarity of those same texture pairs. Each point represents the mean value for one pair of textures. Z-scored timing correlation represents the max cross-correlation of a pair of textures, z-scored across all texture pairs for a given cell, and averaged across all PC-like cortical cells. **B** Mean squared error (MSE) of the prediction of perceived dissimilarity derived from a linear combination of rate and timing differences of increasing-size populations of SA1-like, PC-like, and RA-like cells. Red and teal lines indicates a three-factor model that includes either rate (red) or timing (teal) with each factor in the model derived from the mean across subpopulations of each submodality type (PC-like, RA-like, and SA1-like). The black line indicates a six-factor model including rate and timing from PC-like, SA1-like, and RA-like subpopulations. **C** Accuracy of the complete model (average $n = 10$ population response) in predicting perceived dissimilarity. Each circle represents one texture pair (78 unique pairs, rated by 10 human subjects).

precision[1]. We can thus assess the informativeness of this temporal patterning at each stage of processing and the timescale at which stimulus information is carried. We have previously shown that temporal patterning in afferent responses conveys information about the frequency composition of skin vibrations[3] and, by extension, about textures, particularly fine textures[21]. The temporal precision varies across afferent classes: PC responses are most acute, on the order of 1–5 ms, RA responses exhibit intermediate precision, around 5 ms, and SA1 responses carry little information in their timing. In the nerve, information about vibratory frequency and fine texture is not carried in the firing rates and the perception of vibration and texture cannot be accounted for based solely on rates[2,3,21,36].

The role of timing in cortex has been more controversial because cortical neurons exhibit much greater heterogeneity in their firing rate properties than do nerve fibers[22]. As a result, rate codes are far harder to disambiguate from temporal ones in cortex than in the nerve. Indeed, the responses of individual cortical neurons can be described as temporal filters—each implementing an idiosyncratic temporal differentiation operation on the afferent input, which effectively signal the presence of specific temporal patterns in the input[6]. As a result, some of the information carried in the timing is converted into a rate-based code in this neural population. The question is whether the remaining temporal patterning is still informative and behaviorally relevant. A case in point of this ambiguity is in the coding of vibratory frequency in somatosensory cortex. Cortical neurons have long been known to exhibit robust phase locking to vibrations[5,37,38] and this phase locking was hypothesized to encode vibratory frequency, particularly at the low frequencies (<50 Hz)[37]. However, firing rate was shown to also systematically covary with frequency over this range, which argued for a rate code[38]. On the other hand, the relationship between frequency and rate breaks down at the high frequencies because individual neurons can no longer fire on each stimulus cycle[5]. At those frequencies, then, frequency information seems to be carried solely in the timing.

Vibratory frequency per se is not an ecologically relevant stimulus quantity. One might therefore argue that the temporal code for frequency in somatosensory cortex is an artifact of the highly contrived laboratory stimulus (vibrations delivered by a

motor) and associated sensory task (frequency discrimination). The ethological relevance of this result might then be called into question. In the present study, we show that millisecond-level temporal patterning in the cortical response carries information about more naturalistic stimuli, namely textures, as has been shown in the barrel cortex of rodents[14,39].

**Temporal resolution depends on afferent input**. As a rule, individual neurons in somatosensory cortex receive convergent input from multiple tactile submodalities[6,27,28]. Given that SA1, RA, and PC fibers exhibit different response properties[25], neurons that receive dominant input from any one population of nerve fibers are liable to inherit these differences. As mentioned above, nerve fibers differ in the degree to which their spiking responses to a repeated stimulus are alike. PC responses are the most temporally precise, SA1 responses the least, and RA responses intermediate between these two[3,21]. Here, we show that PC-like cortical neurons consistently carry texture information at the highest temporal resolution (median = 1 ms), RA-like neurons at a lower resolution (3.5 ms), and SA1-like neurons carry little information in their timing, mirroring their afferent counterparts. Interestingly, the temporal precision and informativeness of PC fibers and of PC-like cortical neurons are nearly indistinguishable. One might expect that the three intervening synaptic passes would degrade the response timing and thus its informativeness. However, the manner in which PC input drives cortical activity has been shown to be well suited to preserve timing, by exerting a brief excitatory influence closely followed by a brief inhibitory one[6]. In contrast, RA input is integrated over longer time windows, which leads to a blurring of the temporal patterning, and accounts for the observation that the temporal patterning in the responses of RA-like cortical neurons is less informative than is the temporal patterning in RA afferent responses.

**Temporal patterns scale systematically with speed**. A central question in neuroscience is how we achieve robust object representations despite changing sensory input. Texture perception constitutes an example of this phenomenon. Indeed, afferent responses scale systematically—both in their patterning and in their strength—with changes in scanning speed[21]. Despite the

speed dependence of the texture signal emanating from the periphery, our perception of texture is almost completely independent of scanning speed, even if the texture is scanned passively across the skin[29–31,40]. Scanning speed-related signals in the nerve must thus somehow be discounted in the computation of texture. This process is not trivial, as evidenced by the fact that the converse is not true: Tactile speed perception is highly dependent on texture[40,41].

Rate-based texture signals in cortex have been shown to be more robust to changes in scanning speed than are their counterparts in the nerve[23,42]. This increased robustness to speed could in part be attributed to computations applied to the afferent input and reflected in the output of cortical neurons. Specifically, populations of simulated cortical neurons that reflected temporal and spatial differentiations of their afferent input exhibited responses that were less speed-dependent than was the afferent input itself[23]. Here, we show that this increase in robustness of a rate-based code is complemented by the propagation of the temporal code, which itself scales systematically with speed. One possibility is that this temporal patterning is further converted into a rate-based signal through subsequent differentiation operations. According to this hypothesis, downstream neurons —in secondary somatosensory cortex, e.g.—would carry even more texture information in their rates and this rate-based representation would be even more invariant to changes in speed.

**Temporal patterning of the cortical response shapes texture perception**. Whether temporal patterning in cortical responses shapes perception has proven difficult to establish. Probably the most compelling evidence is that information about the frequency of high-frequency vibrations is carried in the temporal patterning and not the rates; yet their frequency is still discriminable[5]. The link between perception and neural response is thus circumstantial. Here, we provide a more explicit link between temporal patterning and perception: perceptual ratings are better predicted by temporal patterning than by rates.

Furthermore, rate and temporal codes are often put in opposition, where one purportedly drives behavior and the other does not. Our results imply a synergistic integration of the two types of codes: perceived texture is best predicted from the combination of spike rate and timing. This integrative neural code reflects the two codes that carry texture signals in the peripheral nerves: Spatial and temporal. In cortex, texture information carried in spatial patterns of activation of SA1 and RA fibers has been converted into a rate code by the spatial filtering properties of subpopulations of cortical neurons[22,43,44], which individually act as detectors of spatial features in the input. Texture information carried in temporal patterns of activation RA and PC fibers is converted into a rate code by temporal filters[6], which act as detectors of temporal features in the input. As one ascends the somatosensory neuraxis, successive temporal differentiations convert temporal patterns into rate-based signals, but this process is not complete in somatosensory cortex (SC) as evidenced by the residual informativeness of temporal patterning and its ability to predict perception above and beyond rate. Importantly, spatial and temporal codes coexist in SC to encode texture, as seems to be the case in barrel cortex[14].

## Conclusions

Temporal spiking patterns in somatosensory cortex are precise and informative about texture identity. Cells vary in the degree to which texture information is carried in their temporal patterning and this heterogeneity is in part driven by the afferent input: Neurons that receive dominant input from PC fibers, which carry

textural information in their spike timing, themselves carry information in their spike timing. Information carried in the temporal spiking patterns in cortex complements that carried in firing rates. The combination of rate and timing is more predictive of texture perception than is rate or timing in isolation.

## Methods

The cortical responses and human psychophysical data have been described in two previous studies[22,23] and the peripheral nerve responses have been described in another two studies[21,26].

**Animals**. All experimental procedures involving animals were approved by the University of Chicago Institutional Animal Care and Use Committee (ACUP 72042). Cortical data were obtained from two Rhesus macaques (male *Macaca mulatta*, 6–8 years old and 8–11 kg), instrumented with a custom head-post to immobilize the head (for eye-tracking and stable neurophysiological recordings) and a 22-mm wide recording chamber centered on the hand representation in anterior parietal cortex. During training and data collection, animals performed a visual contrast discrimination task so that they would remain awake and calm. In brief, animals fixated on a go-target to initiate the trial, and two circles appeared on the computer monitor. The animal made a saccade to the brighter target to obtain a juice reward. Peripheral responses were collected from 6 anesthetized rhesus macaques as previously described[21].

**Neurophysiology**. Cortical responses: Procedures have been previously described in detail[22,23]. In brief, extracellular recordings were obtained using tungsten electrodes (Epoxylite insulated probes, FHC Inc.) driven into somatosensory cortex— Brodmann's areas 3b, 1, and 2—using a computer-controlled microdrive (NAN Instruments, Nazaret Illit, Israel). We collected the responses from neurons whose receptive fields were centered on the distal fingerpads of digits 2–5. A full recording session (59 textures, presented 5 times at 80 mm/s) lasted at least 30 min and we only report the responses of neurons whose action potential waveforms remained stable over the entire session. For a subset of 49 highly stable neurons, we also collected responses to 10 textures at three additional speeds (60 mm/s, 100 mm/s, and 120 mm/s,[23]).

Peripheral responses: These standard procedures have been previously described in detail[4,21]. In brief, we obtained extracellular recordings of 17 SA1, 15 RA, and 7 PC fibers from the median and ulnar nerves of six Rhesus macaques under isoflurane anesthesia.

**Tactile stimuli**. We presented a diverse set of textures at a controlled speed (80 ± 0.1 mm/s) and force (25 ± 10 g) using a custom-built and designed texture-drum stimulator[21] for both the peripheral and cortical experiments with a different but overlapping texture set. The cortical texture set included 59 textures, including furs, fabrics, papers, and 3D printed gratings and dot patterns. These textures were selected to include features that span spatial scales. The peripheral texture set included 55 textures, 24 of which overlap with the cortical set (Supplemental Table 1). Textures were presented once each in pseudo-random order in each of 5 blocks and the order of presentation changed from block to block. Each texture was presented for 2 s but we only analyzed 500 ms of the steady-state response (excluding the onset and offset transients). Texture presentations were separated by at least 3 seconds, both to prevent neural adaptation and to allow the drum time to shift between textures.

**Human psychophysics**

*Dissimilarity ratings*. These psychophysical ratings have been previously reported[22] and all experiments were approved by the University of Chicago Institutional Review Board (IRB 15-1670). Ten human subjects (10 female, ages 19-24) rated the dissimilarity between pairs of textures. The stimulus set included 13 textures (Supplemental Table 1), yielding 78 unique pairs, and each texture pair was presented 5 times to each subject. If a pair was perceived as identical, the subjects ascribed it a rating of zero. If one pair of textures was perceived as being twice as different as another, the former was to be ascribed a number that was twice as large as that ascribed to the latter. Subjects were free to use the range and were encouraged to use fractions and decimals. Ratings obtained from each subject were normalized by the mean rating across textures in each session and then averaged across sessions.

*Roughness ratings*. These psychophysical ratings have been previously described in two reports, and all experiments were approved by the University of Chicago Institutional Review Board (IRB 15-1670)[22,23]. In brief, six human subjects (5 male, 1 female, ages 18–24) freely rated the perceived roughness of each of the 59 textures used in the neurophysiology experiments. Textures were presented at the same speed and force as those in the neurophysiology recordings (80 mm/s, 25 ± 10 g) and each texture was presented once in each of six experimental blocks. Ratings

were then normalized by the mean rating of each block and averaged across blocks for each subject. The ratings presented here are the average across all subjects.

**Data analysis**. Neurophysiological recordings yielded spike times spanning a period beginning 1 s before to 3 s after skin made contact with a given texture. For all analyses, we used a 500-ms epoch that began 100 ms after the texture contact to exclude transient responses.

*Firing rate-based classification*. We calculated each neuron's firing rate to each texture presentation (141 cells × 59 textures × 5 repeats) by counting the number of spikes during the aforementioned 500-ms window. To assess how informative about texture these firing rates were, we performed a nearest-neighbor classification. For single-cell classification, we calculated the Euclidian distance between one test trial (one repeat of a given texture) and the mean response across all remaining repeats of every texture in the dataset. The classification was correct if the training texture with the lowest distance was the texture presented in the test trial.

*PSTH and cross-correlation classification*. We calculated peri-stimulus time histograms (PSTHs) by convolving spike trains with Gaussian kernels of varying widths (ranging from 0.1 to 500 ms) to assess decoding performance at various temporal precisions. Importantly, all PSTHs were demeaned to remove rate-based information from the signal. All that remained was the waxing and waning of the response in time—a purely time-varying signal that was not confounded by rate. By convolving neural spike trains with Gaussian kernels of increasing width, we obscured increasing amounts of fine timing information. To the extent that this fine timing information is just noise, wider Gaussian kernels should de-noise the signal and improve classification performance. Alternatively, if fine timing is informative, obscuring it should degrade classification. To test this, we performed an analogous classification to the firing rate classification described above. We computed the cross-correlation (using Matlab xcov function, as xcov is a zero-mean cross-correlation) of one test response (500 ms PSTH, computed at a given resolution for one repeat of a given texture) and all other repeats of every texture. To accommodate the fact that responses may be shifted slightly in phase from one repeat to the next, we selected the max cross-correlation across time. We then averaged the max cross-correlation values across all repeats for a given texture, and the classifier selected the training texture that yielded the highest cross-correlation. The classification was correct if the training texture with the highest correlation was the texture presented in the test trial.

*Spike distance*. We sought to quantify the dissimilarity between responses of a given neuron to repeated presentations of a stimulus. To this end, we employed a commonly used method that assigns a minimum cost to transforming one spike train into another[24]. The cost adding or removing a spike is 1, and the cost of moving a spike in time is $q \mid dt \mid$, where $q$ is a parameter. When $q = 0$, there is no penalty to moving spikes in time, such that the minimum cost is simply the difference in spike count. For high values of $q$, spikes that are displaced in time incur a cost. Using this distance metric, we obtained a measure of dissimilarity that is influenced by both the firing rate and temporal patterning of two spike trains.

*Combined rate and timing classification*. To combine rate difference and timing correlation to decode texture identity, we z-scored the distance matrices for each neuron and coding scheme, and multiplied all PSTH cross-correlation values by −1 (to invert similarity into a dissimilarity metric). We then computed the optimal weighted averages of these z-scored distance matrices (rate & timing) for each neuron. To find the optimal weighting, we performed the classification using weights that ranged from 0% rate (100% timing) to 100% rate (0% timing) and identified the weighting that yielded the best performance. To the extent that timing does not improve rate classification, this weighted average would include 100% rate and 0% timing. To the extent that a combination of both coding schemes improves classification performance, the weights would be intermediate. We report the classification performance of the optimal combination, which, in all cases, was an intermediate combination of rate and timing.

*Population analyses*. For the population analyses, we averaged the z-scored distance matrices of n randomly sampled neurons (without replacement). We then performed the classification using this mean distance matrix. For PSTH classification, we used each neuron's optimal decoding resolution. For rate classification, this population analysis did not require the selection of a temporal resolution, as we assessed rate across the entire 500-ms window. For combined rate and timing classification, we weighted each neuron's timing- and rate-based distance matrices (as described in the previous section) and averaged these combined distance matrices across the neuronal population to perform nearest-neighbor classification.

*Homogenous Poisson neurons*. To validate the timing-based classification approach, we simulated Poisson spike trains generated from the measured firing rates. Simulated responses had the same rate as their measured counterparts but lacked temporal fidelity. To generate these model spike trains, we generated a vector of

spike times from a stationary Poisson process where the probability of a spike in each bin (d$t$ = 0.1 ms) was $P = r*dt$, where $r$ is the measured spike rate.

*Jittered spike trains*. At the other extreme of a purely Poisson model is a deterministic response that does not vary across repeats. To simulate responses at predetermined levels of jitter, we took the response of a neuron to one presentation of each texture and jittered each spike time by a specific amount. Specifically, we introduced jitter probabilistically using a Gaussian distribution with mean 0 and a standard deviation defined by the jitter level (between 0.1 and 200 ms). We then added and removed spikes as necessary to ensure that the simulated responses were rate matched with their measured counterparts, given the trial-to-trial variations in the latter. We report the results of generating simulated responses based on the trial with the largest number of spikes and removing spikes of individual simulated responses to match the spike count of each measured response. When we adopted the converse approach—simulating responses from the trial with the lowest number of spikes and adding spikes to match measured spike counts, we obtained nearly identical temporal resolutions (Kolmogorov–Smirnov test, $D = 0.06$, $p = 0.93$; Supplemental Fig. 1).

*Comparing measured spike trains to jittered spike trains (repeatability analysis)*. To quantify how measured cortical responses compared to their simulated jittered counterparts, we used spike distance (described above) to calculate the dissimilarity between the responses evoked by multiple repetitions of the same stimulus at a very high temporal resolution (2 ms, $q = 500$). For each neuron, we first calculated the mean pairwise distance between repeats (for a given texture) and the mean pairwise distance between rate-matched simulated (jittered) responses. Specifically, we calculated the spike distance between every unique combination of responses of one neuron to each texture (repetitions 1 and 2, 1 and 3, 2 and 3, etc.). We then repeated this analysis on the simulated response for each level of jitter. We then identified the jitter levels adjacent to the neuron's cross-repetition mean distance and averaged these two jitter values to obtain an estimate of that neuron's precision. For example, if a neuron's mean cross-repetition spike distance was 15, and the corresponding simulated responses with 1-ms jitter yielded a distance of 13 and the simulated responses with 2-ms jitter yielded a distance of 21, the neuron's resolution was estimated to be 1.5 ms. For each neuron, we repeated this analysis for each texture and used the median of the resulting distribution as the neuron's temporal resolution.

*Submodality composition of cortical cells*. As reported previously[22], we estimated the submodality composition of a neuron's peripheral inputs by performing a multivariate regression of that neuron's z-scored mean rate responses onto the z-scored responses of SA1, RA, and PC afferents to a shared set of 24 textures. We then classified cortical cells with a normalized regression weight of > 0.8 as receiving dominant input from the corresponding population of nerve fibers (SA1, RA, or PC). This criterion yielded 12 PC-like cortical cells ($n = 10$ in area 1, $n = 2$ in area 2), 25 SA-like cortical cells ($n = 9$ in area 3b, $n = 14$ in area 1, $n = 2$ in area 2), and 12 RA-like cortical cells ($n = 2$ in area 3b, $n = 9$ in area 1, $n = 1$ in area 2). Note that this approach to determining the submodality composition of each neuron's input did not take temporal patterning into consideration.

*Cross-speed classification*. To convert neural responses to spikes/mm from spikes/ms, we multiplied spike times by the scanning speed (60, 80, 100, or 120 mm/s). This resulted in warped spike trains that could be compared across speeds. Within-speed classification was performed using a leave-one out procedure. Cross-speed classification was performed in the same way, except that classifiers were trained on four warped repetitions at one speed (80 mm/s, e.g.), and tested on one left-out warped repetition at a different speed (120 mm/s, e.g.). All other methods were as described above.

*Predicting dissimilarity ratings*. We predicted perceived dissimilarity using cross-validated multiple regression. We fit the regression using all but one texture pair and tested it on the left-out texture pair, iterating through each texture pair. We then calculated the mean squared error of this prediction across each of the 78 texture pairs.

*Statistical analyses*. We report a variety of non-parametric statistical tests. To compare unpaired samples from two groups, we used a Mann–Whitney U test (using MATLAB *ranksum* function). When the sample size was smaller than 20 observations, we reported the exact U statistic. Otherwise, we reported the approximated z-statistic. For paired samples from two groups, we used a Wilcoxon signed-rank test (MATLAB *signrank* function). Here too, for sample sizes greater than 20, we reported the approximated z-statistic. When comparing 3 or more groups, we used a one-factor Kruskal–Wallis test (MATLAB *kruskalwallis* function) or a repeated measures Friedman test (MATLAB *friedman* function) and reported the $\chi^2$. To compare the population classification results, we performed a simple 1-sided permutation test, in which we iteratively (10,000 repeats) shuffled data from two samples and identified the proportion of shuffles whose difference was greater than or equal to that of the unshuffled data. We then reported this proportion as the $p$-value. Lastly, to compare empirical cumulative distribution functions from two samples, we used a Kolmogorov–Smirnov test (MATLAB *kstest2* function).

**Reporting summary**. Further information on research design is available in the Nature Research Reporting Summary linked to this article.

## Data availability

The neurophysiological and psychophysical data are available as MATLAB (.mat) data structures in the Source Data folder on GitHub (https://github.com/kthlong/TimingAnalyses). Source data are provided with this paper.

## Code availability

All analyses were performed using MATLAB (R2019b), and all code can be found on GitHub (https://github.com/kthlong/TimingAnalyses).

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

## Acknowledgements

This work was supported by NINDS grants R01 NS101325 (SJB), R35 NS122333 (SJB), and F31 NS110402 (KHL).

## Author contributions

S.J.B. and J.D.L. designed the cortical and psychophysical experiments. J.D.L. collected and organized the cortical neurophysiological data and the human psychophysical data. KHL wrote the code, analyzed the data, and prepared the figures. K.H.L. and S.J.B. wrote the manuscript with feedback from J.D.L.

## Competing interests

The authors declare no competing interests.
