## [Peer Review File · Nature Communications]

Texture is encoded in precise temporal spiking patterns in primate somatosensory cortexREVIEWER COMMENTS

Reviewer #1 (Remarks to the Author):

The significant and exciting finding of this paper is that the timing of neural events in somatosensory cortex maintains information about rich tactile stimuli delivered to the fingertips, and the demonstration that this information is likely contributing to perceptual experience together with mean spike rates. This has proven a difficult area to study, and although similar ideas were first proposed in the 1960's, this is the first paper to provide compelling evidence for a role of timing information in naturalistic tactile experience. As such, this represents a significant contribution to ending the 'battle' between advocates of rate and timing codes, by first demonstrating the validity of timing coding, and then proposing a bridge whereby both codes may contribute to perception. The methodological innovations in analysis described in this paper to quantify the salience of neural timing are significant and likely to be of great interest to neuroscience researchers across a range of disciplines. This is a landmark paper of the highest quality and of broad interest.

The experiments described are carefully controlled. Data from these experiments has been presented in previous publications (Lieber & Bensmaia), however this analysis and interpretation is completely new and of far broader relevance than the previous publications. The data and analyses are thoroughly presented, and I congratulate the authors on the extensive supplementary data figures which help to provide a fuller perspective on the data and the validity of the analyses.

Following are some questions where the paper is unclear or some detail appears to be lacking.

1. The methods do not describe the randomisation of presentation of the textures. Were the 5 repeats of each texture presented sequentially (and with what delay) or were the textures presented in a pseudo-random order?
2. Methods. "Comparing real data to jittered spike trains: For each neuron, we first calculated the mean pairwise distance between repeats (across all textures)..." Please clarify if the mean pairwise distance for repeats was based on rate matched repeats, and if so how this was done.
3. Repeatability analysis doesn't seem to be explained anywhere except Sup Fig 2 which isn't referred to in the main text. Not clear what is being shown here.
4. Supp Fig 6 (and 7A & 7B). It is difficult to compare the performance of cortical PC-like cells with peripheral PC afferents, in part because of the different population sizes which impact the x-axis. Is there any way this can be corrected for in the analysis? It seems surprising that the cortical cells appear to do better with the timing-based classification than the afferents given the jitter added in crossing several synapses. Is this simply an issue of the sampling, or do the authors believe this is a real effect which then merits further discussion?
5. Supp fig 11. Presumably panels C-F are also "perceptual" roughness or dissimilarity. The Methods only describe psychophysics for dissimilarity, but it appears you are also reporting on psychophysical data for roughness: "Consistent with this hypothesis, we found that differences in population firing rate accounted for around 20% of the variance in the dissimilarity rating, as did differences in roughness ratings (Supplemental Figure 11B,C)."
6. Timing correlation (Methods and Supp figures 4, 11, 13). Is this the same as the averaged max cross-correlation value? Can this be normalised to run from -1 to +1 which would make the graphs in Supp Fig 11 & 13 much easier to interpret? It would be useful to understand how much of the correlation in Supp Fig 13 is cell dependent vs texture dependent: this might be achievable with a different-coloured dot per cell given that there are not many PC afferents and PC-like units.

Minor comments.

1. I think Fig 1B might be clearer combined as one panel, as the repetition of the measured data in each panel, and lack of explanation of the jitter levels in the legend, is a little confusing.
2. Typo: In contrast, warping spike trains into spatial units decreased the performance of rate

classifiers by overcompensating for the speed-dependent modulation of >>>speed<<< (Figure 5C).

3. 'SC' appears in the last paragraph of the Discussion and is undefined but is presumably somatosensory cortex which should be spelled out.

4. Several of the supplementary figures are not referred to in the paper - I am not sure what the journal policy is around this.

Richard Vickery

Reviewer #2 (Remarks to the Author):

In their manuscript entitled "Texture is encoded in precise temporal spiking patterns in primate somatosensory cortex" Sliman Bensmaia and colleagues present an interesting and timely study about how surface textures are represented at different levels of the ascending pathway. This is in principle a very exciting topic, which addresses a series of important questions. However, there are multiple aspects that need to be clarified before this manuscript can be fully assessed. Specifically:

Major concerns:

There are no statistics being performed to support many conclusions (nearly in all figures).

The grouping method (PC or SA1) is not well documented. It might be better to delineate the contribution of different afferent types in individual neurons, instead of arbitrarily assigning them to two groups. RA (Meissner) seem to play an important role in texture perception but are largely neglected in the analysis. A full comparison of the different classes would complete the study.

There seems to be many interesting differences between cortex and periphery (Sup Fig 6 and Sup Fig 7); however, the detailed comparison is missing. Why? For example, in general, classification performance is lower in cortex, but the timing-based classification from PC-like cells only is actually better than the periphery.

Minor comments:

Sup Fig 1 needs to be added into the main figure, otherwise it is difficult to understand this novel analysis.

It's still very challenging to understand the details of the timing-based classification. Some visualization like Sup Fig 1 is needed. Please expand.

In the title of Figure 2B, "(n=1)" is confusing. These were multiple experiments?

"rapidly-adapting afferents" should be stated for the first time mentioning "RA".

In the Method for "PSTH and cross-correlation classification", shouldn't the widths of Gaussian kernels be ranging from 0.1 (instead of 1) to 500 ms?

Supplemental Figure 2 was not referred to in the text.

In Sup Fig 5, X-Y axis should be marked better.

Reviewer #3 (Remarks to the Author):

This is an interesting report that carries on from 2 previous publications from this group, analysing

the contribution of temporal and rate coding of surface texture in neurons in the cutaneous hand representation of primary somatosensory cortex (S1). It is generally well presented apart from the fact that 5 supplemental figures are not cited in the text (Supplemental Figures 2, 5, 10, 12 and 13). These should be eliminated if not essential. If they are important, then they should be cited in the main text.

Major

1. My major criticism is related to the very focused view of cortical encoding of tactile texture, concentrating on the S1 neurons whose discharge is most similar to either Pacinian corpuscles (PC) or slowly adapting type 1 receptors (SA). This approach ignores the undoubted contribution of rapidly adapting (RA) receptors to texture encoding. RA mechanoreceptors make up the largest proportion of low threshold cutaneous mechanoreceptors innervating the human fingertips (Johansson and Vallbo 1979) – approximately 140 units/cm² of skin as compared to 70 units/cm² for SAI units and only 20 units/cm² for PC afferents. RA inputs are ignored in this report, and in previous work from this laboratory. Inspection of the supplemental figures suggests that this group encodes temporal inputs at least as well as PC afferents and sometimes better (Suppl Fig. 7B). Concentrating upon the extremes of the cortical population encoding texture gives an incomplete picture. What would happen if the analyses presented in Fig. 3 were separated into PC-like, SA-like and “other” (presumably mainly RA), rather than pooling all data under “all cells”? We get a hint of an answer in Fig. 4A since there were no PC-like neurons recorded in area 3b, and the neurons here have higher correct classifications than neurons in area 1 where the majority of PC-like units were found. In a similar vein, the analysis presented in Suppl Fig. 6CD should be extended to include the 104 neurons that were excluded from this analysis: where do they fall between the extremes of timing and rate classification? The authors could address these issues by reporting, separately, the results of the large group of excluded cells. Certainly, the plots of “all cells” are contaminated by the presence of the SA and PC-like units (e.g. Fig. 3AB) so that one never has any appreciation of the relative resolution of the excluded cells. I believe that this manuscript would be significantly strengthened with a more balanced presentation of the results. At the very least, explicit data on the other cells must be included in the supplemental data and referred to in the main text.

2. Temporal precision decreases at successive stages of cortical processing (ln239-254): I am confused here – inspection of Suppl Fig. 8 shows that there were no PC-like neurons recorded in area 3b, and yet the claim is made that there is better spike-timing in area 3b than in areas 1 (where 10/12 PC-like neurons were recorded) or 2 (2/12), Fig. 4. This leaves the other neurons, SA1-like and presumed RA-like, responsible for spike-timing, and not PC-like neurons. The authors acknowledge this problem but provide no explanation. Finally, the suggestion that temporal precision decreases on going from area 3b to more caudal regions needs statistical support (none provided). The authors must fully address these issues in the main text.

Specific points

1. Fig. 1 is particularly important but is not explained sufficiently. Please indicate the classification for each cell plotted in Fig. 1A (SA, PC). What do the plots in Fig.1B show? Which neuron is this – one of those illustrated in A or some other cell? What do the colors represent here, and why are there 2 curves?

2. The corresponding Suppl Fig.1 is equally poorly explained. The legend should explain that you took trial #x (please identify) and then applied a jitter. My question – if you chose another trial, were the results similar, and to what extent was this true for the population of cells analysed, particularly as a function of the classification of the cells?

3. Population comparisons. There are many statements to the effect that some categories of neurons or certain neural codes are “better” than others (Fig. 2C, legend; Fig. 3 legend; Ln 230; Ln 248; Ln 304; ln643) but there are never any statistical results provided to support such statements. The authors must either provide statistical support for these statements or remove them.

4. Temporal coding: peripheral vs cortical. It seems that this result- similarity between PC afferents and PC-like cortical cells - is simply a function of the method used to categorize and retain cortical cells for analysis. Basically, the cortical units had to behave similarly to primary afferents (PC or SA1). The similarity in temporal coding (peripheral vs cortical) reflects the restricted sampling of cortical neurons.

5. Discussion, ln374-383. The authors speculate that downstream neurons, possible in S2, would carry more texture information in their rates, independent of speed. In fact, this coding mechanism is present much earlier in the cortical processing of tactile inputs. Bourgeon et al (2016) showed that the firing rates of many neurons in area 3b, and also area 1, covary with tactile roughness independent of speed. This work should have been cited and must be discussed here.

Minor

Human psychophysical data: were these data already published in Lieber and Bensmaia (2019)? If yes, then this should be stated in the paper.

Suppl Fig.1: A. Need time scale. Also, please report the spike distance for the trial of real data that was used for the "jitter" analysis (in the legend). B. Please explain what the two curves represent.

Ln125. Is this the correct reference, De Lafuente & Romo 2006? This paper did not use a spike distance metric.

Ln126-7. It seems counterintuitive that a small jitter would drive large dissimilarity values. Suppl Fig 1 shows smaller spike distances for smaller jitters. This should be explained in the text.

Fig. 2A. Were all textures used for this plot? Please specify in the legend.

Fig 2B. What was the categorization of this cell? It does not appear to be particularly representative of the population response as shown in Fig. 2C. The authors should choose another, more representative, example. This cell is also used in Suppl Fig. 4A: same suggestion.

Submodality analysis (results, methods). It would be helpful if the authors defined what they mean by the "response" of each neuron or peripheral afferent, as applied to the regression analyses.

Ln211-217: I realize that the analysis is described in the Methods (Ln452-7) but the authors need to explain here how they determined that the spiking patterns were more "informative". In relation to the same point, the difference between PC and SA1-like neurons in Fig. 3B is not obvious. This is another example where a statistical comparison of the two distributions is needed to support this suggestion; in the absence of any difference, then the statement should be removed.

Supplemental Fig. 6: I would not describe a sample of 7 PC afferents as a population. The term "population" must be removed from the title (B).

Ln219: it would be better to refer to the "peripheral afferents" rather than the "nerves".

Ln 279-81: This statement needs statistical support (higher level with both codes vs rate or timing alone).

Fig. 6A. The title refers to timing similarity while the x-axis label refers to timing correlation. Which term is correct?

Methods. Were the individual textures presented in blocks of trials, or were the various textures presented in a pseudo random order? Please clarify.

Suppl Table 1. Why does this list of textures include many that were not used in any of the recordings. I suggest omitting these.

Ln372-3. This observation (smooth vs textured) was first made by Depeault et al. (2008). Moreover, the results are complex: when roughness was systematically changed while keeping the material properties of the surfaces identical, a potential confound in this study, then surfaces were rated as moving slower for the roughest surface. It is likely that the smoothest surface tested here (chiffon) was ineffective in activating peripheral afferents. The text must be modified to include explicit recognition and discussion of the work by Depeault et al.

Suppl Fig 11: The title over each plot is incorrect (y is plotted as a function of x, and not vice-versa as here). Please change the order for each title. This same issue also arises for Fig. 6A. Further, what is the difference between perceptual roughness and perceptual dissimilarity? Finally, the legend should specify numbers of subjects and texture pairs contributing to the plots.

We would like to thank the reviewers for their kind words and insightful comments. We have made substantial changes to the manuscript, detailed below. In brief, we added statistics throughout, systematically reported the results of analyses carried out on neurons that did not fall into the PC-like or SA1-like categories, considerably amplified the Methods section, and included discussion of certain peculiarities in the findings. We feel the paper is much improved as a result of these changes and hope you will too.

REVIEWER COMMENTS

Reviewer #1 (Remarks to the Author):

The significant and exciting finding of this paper is that the timing of neural events in somatosensory cortex maintains information about rich tactile stimuli delivered to the fingertips, and the demonstration that this information is likely contributing to perceptual experience together with mean spike rates. This has proven a difficult area to study, and although similar ideas were first proposed in the 1960's, this is the first paper to provide compelling evidence for a role of timing information in naturalistic tactile experience. As such, this represents a significant contribution to ending the 'battle' between advocates of rate and timing codes, by first demonstrating the validity of timing coding, and then proposing a bridge whereby both codes may contribute to perception. The methodological innovations in analysis described in this paper to quantify the salience of neural timing are significant and likely to be of great interest to neuroscience researchers across a range of disciplines. This is a landmark paper of the highest quality and of broad interest.

The experiments described are carefully controlled. Data from these experiments has been presented in previous publications (Lieber & Bensmaia), however this analysis and interpretation is completely new and of far broader relevance than the previous publications. The data and analyses are thoroughly presented, and I congratulate the authors on the extensive supplementary data figures which help to provide a fuller perspective on the data and the validity of the analyses.

Thank you for the kind words!

Following are some questions where the paper is unclear or some detail appears to be lacking.

1. The methods do not describe the randomisation of presentation of the textures. Were the 5 repeats of each texture presented sequentially (and with what delay) or were the textures presented in a pseudo-random order?

We have added the following details to our methods:

"Textures were presented once each in pseudo-random order in each of 5 blocks and the order of presentation changed from block to block. Each texture was presented for 2 seconds but we only analyzed 500 ms of the steady-state response (excluding the onset and offset transients). Texture presentations were separated by at least 3 seconds, both to prevent neural adaptation and to allow the drum time to shift between textures.

2. Methods. "Comparing real data to jittered spike trains: For each neuron, we first calculated the mean pairwise distance between repeats (across all textures)..."

Please clarify if the mean pairwise distance for repeats was based on rate matched repeats, and if so how this was done.

We have expanded on this section to clarify. In brief, we found mean pairwise distance for each model individually (so we never found the distance between a recorded response and a simulated response, for example). Instead, we created rate-matched simulated responses (the jitter models), and we found each jitter model's mean pairwise distance between repeats for a given neuron and texture:

"Comparing measured spike trains to jittered spike trains (repeatability analysis): To quantify how measured cortical responses compared to their simulated jittered counterparts, we used spike distance (described above) to calculate the dissimilarity between the responses evoked by multiple repetitions of the same stimulus at a very high temporal resolution (2ms, $q = 500$). For each neuron, we first calculated the mean pairwise distance between repeats (for a given texture) and the mean pairwise distance between rate-matched simulated (jittered) responses. Specifically, we calculated the spike distance between every unique combination of responses of one neuron to

each texture (repetitions 1 and 2, 1 and 3, 2 and 3, etc.). We then repeated this analysis on the simulated response for each level of jitter. We then identified the jitter levels adjacent to the neuron's cross-repetition mean distance and averaged these two jitter values to obtain an estimate of that neuron's precision. For example, if a neuron's mean cross-repetition spike distance was 15, and the corresponding simulated responses with 1-ms jitter yielded a distance of 13 and the simulated responses with 2-ms jitter yielded a distance of 21, the neuron's resolution was estimated to be 1.5 ms. For each neuron, we repeated this analysis for each texture and used the median of the resulting distribution as the neuron's temporal resolution."

3. Repeatability analysis doesn't seem to be explained anywhere except Sup Fig 2 which isn't referred to in the main text. Not clear what is being shown here.

Thank you for pointing out this oversight – we have clarified by referring to the analyses in Figure 1 (using jittered models to quantify the reliability of recorded responses) as the repeatability analysis. We did so in the text in the first section of the results, as well in the methods section describing the “comparing real data to jittered spike trains”. We have also clarified the legend of Supplemental Figure 2:

“A| Cumulative distribution of optimal resolutions for the repeatability (Figure 1) and timing-based classification (Figure 2) analyses.”

4. Supp Fig 6 (and 7A & 7B). It is difficult to compare the performance of cortical PC-like cells with peripheral PC afferents, in part because of the different population sizes which impact the x-axis. Is there any way this can be corrected for in the analysis? It seems surprising that the cortical cells appear to do better with the timing-based classification than the afferents given the jitter added in crossing several synapses. Is this simply an issue of the sampling, or do the authors believe this is a real effect which then merits further discussion?

Thank you for this feedback; we have plotted these together, such that peripheral and cortical performance can be more readily compared. We have added a passage in the document to speculate as to the origins of this surprising result:

“Surprisingly, the temporal precision and informativeness of PC fibers and of PC-like cortical neurons are nearly indistinguishable. One might expect that the three intervening synaptic passes would degrade the response timing and thus its informativeness. One possibility is that the convergence of PC inputs onto individual cortical neurons (via intervening structures) compensates for any synaptically induced loss of temporal precision as signals propagate along the neuraxis.”

5. Supp fig 11. Presumably panels C-F are also “perceptual” roughness or dissimilarity. The Methods only describe psychophysics for dissimilarity, but it appears you are also reporting on psychophysical data for roughness: “Consistent with this hypothesis, we found that differences in population firing rate accounted for around 20% of the variance in the dissimilarity rating, as did differences in roughness ratings (Supplemental Figure 11B,C).”

We have added the term “perceptual” to Supp Fig 11 and added the following description of the psychophysical procedure:

“Roughness ratings: These psychophysical ratings have been previously described in two reports (Lieber and Bensmaia, 2019, 2020). In brief, six human subjects (5 male, 1 female, ages 18-24) freely rated the perceived roughness of each of the 59 textures used in the neurophysiology experiments. Textures were presented at the same speed and force as those in the neurophysiology recordings (80 mm/s, 25 ± 10 g) and each texture was presented once in each of six experimental blocks. Ratings were then normalized by the mean rating of each block and averaged across blocks for each subject. The ratings presented here are the average across all subjects.”

6. Timing correlation (Methods and Supp figures 4, 11, 13). Is this the same as the averaged max cross-correlation value? Can this be normalised to run from -1 to +1 which would make the graphs in Supp Fig 11 & 13 much easier to interpret? It would be useful to understand how much of the correlation in Supp Fig 13 is cell dependent vs texture dependent: this might be achievable with a different-coloured dot per cell given that there are not many PC afferents and PC-like units.

It is the same as the max cross-correlation. Thanks for pointing out the need for clarity: we have addressed this in two ways. First, we z-scored all max-cross correlation values for a given cell, and plotted the averages across cells

(Figures 6, Supplementary figures 11 and 13). To address your observation about Supplementary Figure 13, we have added an additional panel that illustrates each individual cell's adjusted R^2 value. Plotting each cell on the same plot was too cluttered (78 points x 12 cells), but panel C illustrates that this effect is shared among most cells.

Minor comments.

1. I think Fig 1B might be clearer combined as one panel, as the repetition of the measured data in each panel, and lack of explanation of the jitter levels in the legend, is a little confusing.

This is a very helpful comment, thank you! We removed this panel altogether, and instead added in panels from Supplementary Figure 1. This addresses everything that panel B was meant to convey, but hopefully in a less confusing way.

2. Typo: In contrast, warping spike trains into spatial units decreased the performance of rate classifiers by overcompensating for the speed-dependent modulation of >>>speed<<< (Figure 5C).

Thank you! We replaced "speed" with "firing rates".

3. 'SC' appears in the last paragraph of the Discussion and is undefined but is presumably somatosensory cortex which should be spelled out.

Thank you; we had spelled it out in the Results section, but we have also added the full name in the Discussion:

"As one ascends the somatosensory neuraxis, successive temporal differentiations convert temporal patterns into rate, but this process is not complete in somatosensory cortex (SC) as evidenced by..."

4. Several of the supplementary figures are not referred to in the paper - I am not sure what the journal policy is around this.

Thank you for pointing this out. We have added references to each of these supplemental figures within the text.

Richard Vickery

Reviewer #2 (Remarks to the Author):

In their manuscript entitled “Texture is encoded in precise temporal spiking patterns in primate somatosensory cortex” Sliman Bensmaia and colleagues present an interesting and timely study about how surface textures are represented at different levels of the ascending pathway. This is in principle a very exciting topic, which addresses a series of important questions. However, there are multiple aspects that need to be clarified before this manuscript can be fully assessed.

Specifically:

Major concerns:

There are no statistics being performed to support many conclusions (nearly in all figures).

Excellent point. We have included statistical tests throughout. We describe our statistical analyses in a dedicated section in the Methods:

“Statistical analyses: We report a variety of non-parametric statistical tests. To compare unpaired samples from two groups, we used a Mann-Whitney U test (using MATLAB ranksum function). When the sample size was smaller than 20 observations, we reported the exact U statistic. Otherwise, we reported the approximated z-statistic. For paired samples from two groups, we used a Wilcoxon signed-rank test (MATLAB signrank function). Here too, for sample sizes greater than 20, we reported the approximated z-statistic. When comparing 3 or more groups, we used a one-factor Kruskal-Wallis test (MATLAB kruskalwallis function) or a repeated measures Friedman test (MATLAB friedman function) and reported the χ^2 . To compare the population classification results, we performed a simple 1-sided permutation test, in which we iteratively (10,000 repeats) shuffled data from two samples and identified the proportion of shuffles whose difference was greater than or equal to that of the unshuffled data. We then reported this proportion as the p-value. Lastly, to compare empirical cumulative distribution functions from two samples, we used a Kolmogorov-Smirnov test (MATLAB kstest2 function).”

The grouping method (PC or SA1) is not well documented. It might be better to delineate the contribution of different afferent types in individual neurons, instead of arbitrarily assigning them to two groups. RA (Meissner) seem to play an important role in texture perception but are largely neglected in the analysis. A full comparison of the different classes would complete the study.

We have previously found that, with our particular texture set, neurons fall along a continuum of SA1-like to PC-like. When we evaluated the main axes of dimensionality in both the peripheral and cortical responses, the second dominant source of shared variance in both peripheral and cortical responses was highly correlated ($r = 0.89$), and this principal axis reliably separated SA1 from PC afferents and SA1-like cells from PC-like cells.

We do not have a good way to identify neurons whose dominant input is from RA fibers and distinguish them from neurons that receive a combination of PC and SA1 input. Because of this, we focus specifically on these two extremes.

However, to better represent those cells that are intermediate to SA1-like and PC-like cells, we have updated all figures to include a trace of all “other” cells (neither SA1- nor PC-like). This includes Figure 3 and Supplemental Figures 8 and 9. We also clearly articulate that these two extremes are used for illustrative purposes only. For example:

“Here, we show that PC-like cortical neurons exhibit more reliable temporal patterning and carry more information about texture in their spike timing than do their SA1-like counterparts. As mentioned above, however, these neural populations are at extremes of a continuum: Neurons that receive balanced PC and SA1 input will exhibit intermediate response properties. However, to the extent that they receive PC input, they are liable to carry texture information in the temporal patterning of their response. Consistent with this hypothesis, the optimal decoding resolution is on the order of a few milliseconds, even in many cells that receive dominant SA1 input, despite the low temporal resolution of this dominant input.”

There seems to be many interesting differences between cortex and periphery (Sup Fig 6 and Sup Fig 7); however, the detailed comparison is missing. Why? For example, in general, classification performance is lower in cortex, but the timing-based classification from PC-like cells only is actually better than the periphery.

The comparison between PC-like cells and PC afferents could be the result of convergence: if a given cortical cell receives convergent input from multiple well-timed afferents, it could rely on synchrony in their responses to de-noise the timing of its outputs. We have not yet created a model for this, but we are working on one. It is worth noting, however, that there is no statistical difference between the timing classification performance in PC-like cortical cells and PC afferents, so although some cells in cortex allow for higher classification (and this effect is certainly very interesting), the populations of cells are indistinguishable.

We have elaborated on a section in the paper describing these differences:

“Next, we compared the temporal coding in cortex to its peripheral counterpart using a shared set of 24 textures (Supplemental Figure 8). We found striking similarities between PC-like cortical cells and PC afferents: they yielded similar classification performance (Mann-Whitney U test: $U = 126.5$, $P = 0.61$) with similar optimal temporal resolutions. Likewise, temporal coding was weak for both SA1 fibers and SA1-like cortical neurons, though spike timing in the former was more informative than in the latter ($U = 366$, $P < 0.0001$). The contrast between the two populations of nerve fibers and their downstream targets was also observed at the population level. In small populations of both PC fibers and PC-like cortical cells, timing classification exceeded rate classification (permutation test comparing rate classification to timing classification in groups of 5 cells, peripheral: $P < 0.01$, cortical: $P < 0.0001$; Figure 3D, Supplemental Figure 9B). In contrast, populations of SA1 fibers and SA1-like cortical cells yielded better classification performance with rate than with timing (permutation test in groups of 5 cells, peripheral: $P < 0.0001$, cortical: $P < 0.0001$; Figure 3C, Supplemental Figure 9A). These results bolster the hypothesis that the temporal coding properties of cortical neurons are inherited from their inputs.”

And to speculate about the high performance of PC-like neurons:

“Surprisingly, the temporal precision and informativeness of PC fibers and of PC-like cortical neurons are nearly indistinguishable. One might expect that the three intervening synaptic passes would degrade the response timing and thus its informativeness. One possibility is that the convergence of PC inputs onto individual cortical neurons (via intervening structures) compensates for any synaptically induced loss of temporal precision as signals propagate along the neuraxis.”

Minor comments:

Sup Fig 1 needs to be added into the main figure, otherwise it is difficult to understand this novel analysis.

This was great advice, thank you. We have included Sup Fig 1 as part of the main Figure 1.

It's still very challenging to understand the details of the timing-based classification. Some visualization like Sup Fig 1 is needed. Please expand.

We have added visualizations to a supplementary figure (Sup Fig 2).

In the title of Figure 2B, “(n=1)” is confusing. These were multiple experiments?

This was all one experiment, but the analyses either focused on individual cells or groups of cells; we were trying to highlight that difference between 2B (single) and 2C (groups). We have removed “n=1” and left it at “single cell classification” and “population classification”.

“rapidly-adapting afferents” should be stated for the first time mentioning “RA”.

Thanks for catching this; we have added the full name when we first mention RAs:

“...slowly-adapting afferents type-1 (SA1) fibers tend to respond to slower, larger skin deflections and, accordingly, reflect coarse textural features in their spatial pattern of activation; rapidly -adapting (RA) fibers exhibit response and texture-coding properties...”

In the Method for “PSTH and cross-correlation classification”, shouldn't the widths of Gaussian kernels be ranging from 0.1 (instead of 1) to 500 ms?

Yes, thank you! We have revised this to say:

"We calculated peri-stimulus time histograms (PSTHs) by convolving spike trains with Gaussian kernels of varying widths (ranging from 0.1 to 500 ms)"

Supplemental Figure 2 was not referred to in the text.

Thanks for pointing this out – we referenced it here (due to some changes, it is now Supplemental Figure 3):

"In agreement with our findings from the repeatability analysis, we found that classification performance peaked at a high temporal resolution (<5 ms) in individual cortical cells (Figure 2A) and resolutions were similar to those found in the repeatability analysis (Supplemental Figure 4)."

In Sup Fig 5, X-Y axis should be marked better.

Thank you, we have added labels!

Reviewer #3 (Remarks to the Author):

This is an interesting report that carries on from 2 previous publications from this group, analysing the contribution of temporal and rate coding of surface texture in neurons in the cutaneous hand representation of primary somatosensory cortex (S1). It is generally well presented apart from the fact that 5 supplemental figures are not cited in the text (Supplemental Figures 2, 5, 10, 12 and 13). These should be eliminated if not essential. If they are important, then they should be cited in the main text.

Major

1. My major criticism is related to the very focused view of cortical encoding of tactile texture, concentrating on the S1 neurons whose discharge is most similar to either Pacinian corpuscles (PC) or slowly adapting type 1 receptors (SA). This approach ignores the undoubted contribution of rapidly adapting (RA) receptors to texture encoding. RA mechanoreceptors make up the largest proportion of low threshold cutaneous mechanoreceptors innervating the human fingertips (Johansson and Vallbo 1979) – approximately 140 units/cm² of skin as compared to 70 units/cm² for SAI units and only 20 units/cm² for PC afferents. RA inputs are ignored in this report, and in previous work from this laboratory. Inspection of the supplemental figures suggests that this group encodes temporal inputs at least as well as PC afferents and sometimes better (Suppl Fig. 7B). Concentrating upon the extremes of the cortical population encoding texture gives an incomplete picture. What would happen if the analyses presented in Fig. 3 were separated into PC-like, SA-like and "other" (presumably mainly RA), rather than pooling all data under "all cells"? We get a hint of an answer in Fig. 4A since there were no PC-like neurons recorded in area 3b, and the neurons here have higher correct classifications than neurons in area 1 where the majority of PC-like units were found. In a similar vein, the analysis presented in Suppl Fig. 6CD should be extended to include the 104 neurons that were excluded from this analysis: where do they fall between the extremes of timing and rate classification? The authors could address these issues by reporting, separately, the results of the large group of excluded cells. Certainly, the plots of "all cells" are contaminated by the presence of the SA and PC-like units (e.g. Fig. 3AB) so that one never has any appreciation of the relative resolution of the excluded cells. I believe that this manuscript would be significantly strengthened with a more balanced presentation of the results. At the very least, explicit data on the other cells must be included in the supplemental data and referred to in the main text.

This is very helpful feedback, and we have made an attempt to present a more comprehensive and balanced view of the role of non SA1- / PC- like neurons.

We have previously found that neurons fall along a continuum of SA1-like to PC-like. When we evaluated the main axes of dimensionality in both the peripheral and cortical responses, the second dominant source of shared variance in both peripheral and cortical responses was highly correlated ($r = 0.89$), and this principal axis reliably separated SA1 from PC afferents and SA1-like cells from PC-like cells.

We do not have a good way to distinguish neurons whose dominant input is from RA fibers from neurons that receive input from PC and SA1 fibers.

However, to better represent those cells that do not fall on the SA1 or PC extremes, we have updated all figures to include a trace of all “other” cells (neither SA1- nor PC-like). This includes Figure 3 and Supplemental Figures 8 and 9.

2. Temporal precision decreases at successive stages of cortical processing (In239-254): I am confused here – inspection of Suppl Fig. 8 shows that there were no PC-like neurons recorded in area 3b, and yet the claim is made that there is better spike-timing in area 3b than in areas 1 (where 10/12 PC-like neurons were recorded) or 2 (2/12), Fig. 4. This leaves the other neurons, SA1-like and presumed RA-like, responsible for spike-timing, and not PC-like neurons. The authors acknowledge this problem but provide no explanation. Finally, the suggestion that temporal precision decreases on going from area 3b to more caudal regions needs statistical support (none provided). The authors must fully address these issues in the main text.

We have added the following explanation, although it is only our current working hypothesis:

“Notably, differences in the prevalence of temporally precise neurons across cortical fields is not driven by differences in submodality input. That is, area 3b contained the most temporally precise neurons despite the fact that none were classified as receiving dominant input from PC fibers (Supplemental Figure 10). In other words, the incidence of PC-like neurons in area 1 was not sufficient to overcome the overall decrement in temporal precision.”

We have provided statistical support for our comparison across areas 3b, 1, and 2:

*“As expected, the preponderance of neurons that carry information about texture in spike timing decreased at successive stages of processing, as evidenced by a decrease in spike-timing based texture classification across areas (Kruskal-Wallis test comparing areas 3b, 1, and 2, chi-square statistic: 9.37, p-value: 9.2e-3; post-hoc Wilcoxon rank-sum test comparing areas 3b and 1, z-statistic: 2.0, p-value: .02, and areas 1 and 2, z-statistic: 1.7, p-value: .04; **Error! Reference source not found.A, Error! Reference source not found.**)”*

Specific points

1. Fig. 1 is particularly important but is not explained sufficiently. Please indicate the classification for each cell plotted in Fig. 1A (SA, PC). What do the plots in Fig.1B show? Which neuron is this – one of those illustrated in A or some other cell? What do the colors represent here, and why are there 2 curves?

Thank you for this feedback – we have implemented the following changes to more thoroughly explain Figure 1.

- (1) Rather than labeling the neurons (most of which are not SA1-like or PC-like), we have added PC-like and SA1-like rasters to Supplemental Figure 7. Because we do not introduce the submodality composition of individual cortical cells until later in the paper, we feel this approach shows the reader some representative responses from these distinct subpopulations without further complicating Figure 1.
 - a. Fig 1 legend: “The bottom row of responses, colored in black, is from the example cell used in B-E of this figure.”
- (2) We indicated which of the rasters in A shows the responses of the example cell plotted in B-E.
- (3) We have removed panel B as this seemed only to confuse, and everything it was meant to convey is better illustrated in the added panels originally from Supp Fig 1.
- (4) We included Supplemental Figure 1 in this figure to illustrate the different stages of this analysis. B-E show the analysis pipeline for one example cell, and F shows the distribution temporal resolutions for the full population.

2. The corresponding Suppl Fig.1 is equally poorly explained. The legend should explain that you took trial #x (please identify) and then applied a jitter. My question – if you chose another trial, were the results similar, and to what extent was this true for the population of cells analysed, particularly as a function of the classification of the cells?

We included Supplemental Fig 1 in with Figure 1, and we have added more thorough descriptions, including a mention that, in this particular example (this cell and this texture), we used the second trial, which we have also labelled with an asterisk in the figure itself:

“Response of one example neuron to 5 repeated presentations of one example texture (an upholstery fabric). The asterisk indicates the specific trial response (trial #2) that was used to generate the jitter simulated responses.

Colored rasters represent rate-matched simulated responses with different amounts of jitter. The grey raster is a rate-matched Poisson model. Spike distance values ($q=500$) represent mean pairwise spike distance across the 5 repetitions shown."

Regarding your question about simulating responses with jitter from different trials, we have added the following text to the methods, as well as a supplementary figure:

"Here, we report the results of generating simulated responses based on the trial with the largest number of spikes and removing spikes of individual simulated responses to match the spike count of each measured response. When we adopted the converse approach – simulating responses from the trial with the lowest number of spikes and adding spikes to match measured spike counts –, we obtained nearly identical temporal resolutions (Kolmogorov-Smirnov test, $D = 0.06$, $P = 0.93$; Supplemental Figure 1)"

3. Population comparisons. There are many statements to the effect that some categories of neurons or certain neural codes are "better" than others (Fig. 2C, legend; Fig. 3 legend; Ln 230; Ln 248; Ln 304; Ln643) but there are never any statistical results provided to support such statements. The authors must either provide statistical support for these statements or remove them.

Thank you; we have included statistics support for these statements. We did not add statistics to the paper's figure legends (not including supplemental figures), but we included the relevant statistics in the text.

We also include the following description in the methods of all the statistical tests we used to support our findings:

"Statistical analyses: We report a variety of non-parametric statistical tests. To compare unpaired samples from two groups, we used a Mann-Whitney U test (using MATLAB ranksum function). When the sample size was smaller than 20 observations, we reported the exact U statistic. Otherwise, we reported the approximated z-statistic. For paired samples from two groups, we used a Wilcoxon signed-rank test (MATLAB signrank function). Here too, for sample sizes greater than 20, we reported the approximated z-statistic. When comparing 3 or more groups, we used a one-factor Kruskal-Wallis test (MATLAB kruskalwallis function) or a repeated measures Friedman test (MATLAB friedman function) and reported the χ^2 . To compare the population classification results, we performed a simple 1-sided permutation test, in which we iteratively (10,000 repeats) shuffled data from two samples and identified the proportion of shuffles whose difference was greater than or equal to that of the unshuffled data. We then reported this proportion as the p-value. Lastly, to compare empirical cumulative distribution functions from two samples, we used a Kolmogorov-Smirnov test (MATLAB kstest2 function)."

4. Temporal coding: peripheral vs cortical. It seems that this result- similarity between PC afferents and PC-like cortical cells - is simply a function of the method used to categorize and retain cortical cells for analysis. Basically, the cortical units had to behave similarly to primary afferents (PC or SA1). The similarity in temporal coding (peripheral vs cortical) reflects the restricted sampling of cortical neurons.

The temporal patterning, and the informativeness of temporal codes, were not taken into account when classifying cortical cells as PC- or SA1-like. Instead, cells were labeled as PC-like if their texture-evoked firing rates were similar to PC afferents. We now point this out not only in the Results section, but also in the Methods section.

5. Discussion, Ln374-383. The authors speculate that downstream neurons, possible in S2, would carry more texture information in their rates, independent of speed. In fact, this coding mechanism is present much earlier in the cortical processing of tactile inputs. Bourgeon et al (2016) showed that the firing rates of many neurons in area 3b, and also area 1, covary with tactile roughness independent of speed. This work should have been cited and must be discussed here.

Thank you for pointing out this omission. We now cite the Bourgeon paper and have clarified our statement about downstream neurons, possibly in S2.

"Rate-based texture signals in cortex have been shown to be more robust to changes in scanning speed than are their counterparts in the nerve (Bourgeon et al., 2016; Lieber and Bensmaia, 2020). This increased robustness to speed could in part be attributed to computations applied to the afferent input and reflected in the output of cortical neurons. Specifically, populations of simulated cortical neurons that reflected temporal and spatial differentiations of their afferent input exhibited responses that were less speed-dependent than was the afferent

input itself (Lieber and Bensmaia, 2020). Here, we show that this increase in robustness of a rate-based code is complemented by the propagation of the temporal code, which itself scales systematically with speed. One possibility is that this temporal patterning is further converted into a rate-based signal through successive differentiation operations. According to this hypothesis, downstream neurons – in secondary somatosensory cortex, e.g. – would carry even more texture information in their rates and this rate-based representation would be even more invariant to changes in speed.”

Minor

Human psychophysical data: were these data already published in Lieber and Bensmaia (2019)? If yes, then this should be stated in the paper.

Yes, we now mention that each of these data sets have been previously described in the relevant passages in the Methods.:

Suppl Fig.1: A. Need time scale. Also, please report the spike distance for the trial of real data that was used for the “jitter” analysis (in the legend). B. Please explain what the two curves represent.

We added the time scale – thank you for catching that! We cannot report a spike distance for the trial of real data; spike distance measures the difference between a *pair* of spike trains. We report the spike distance for recorded responses and for each of the jitter models. The two curves represent recorded and Poisson; we have clarified this in the legend, which is now part of Figure 1:

“The black trace is derived from the measured response of the neuron and the grey trace is derived from a rate-matched Poisson model to ‘Ruby Dots’.”

Ln125. Is this the correct reference, De Lafuente & Romo 2006? This paper did not use a spike distance metric.

No; thank you for pointing this out!! We have removed this reference.

Ln126-7. It seems counterintuitive that a small jitter would drive large dissimilarity values. Suppl Fig 1 shows smaller spike distances for smaller jitters. This should be explained in the text.

This was confusing word choice; thank you for pointing this out. We have clarified with the following:

“Varying the cost of shifting spikes in time allows us to manipulate the temporal resolution of this metric: At one larger extreme in cost, even small inconsistencies in the precise timing of spikes drives relatively large dissimilarity values; at the other extreme in cost (close or equal to zero), distance values are driven only by differences in spike count.”

Fig. 2A. Were all textures used for this plot? Please specify in the legend.

Yes, all 59 textures were used. The legend mentioned this near the end of the description of A, but we added it right at the beginning:

“Classification performance (percentage of textures correctly classified from the full texture set, comprising 59 unique textures) is best at high temporal resolutions (1-5 ms).”

Fig 2B. What was the categorization of this cell? It does not appear to be particularly representative of the population response as shown in Fig. 2C. The authors should choose another, more representative, example. This cell is also used in Suppl Fig. 4A: same suggestion.

This cell is a PC-like cortical cell, and is not representative of the average. It is representative of PC-like cortical cells (with peak performance falling in the median of all PC-like cells), but in an effort to better illustrate a representative cell in the general population, we have added another example cell whose peak performance falls in the median of all cortical cells.

Submodality analysis (results, methods). It would be helpful if the authors defined what they mean by the “response” of each neuron or peripheral afferent, as applied to the regression analyses.

We have clarified this:

“As reported previously (Lieber and Bensmaia, 2019), we estimated the submodality composition of a neuron’s peripheral inputs by regressing that neuron’s z-scored mean rate responses onto the z-scored responses of SA1, RA, and PC afferents to a shared set of 24 textures.”

Ln211-217: I realize that the analysis is described in the Methods (Ln452-7) but the authors need to explain here how they determined that the spiking patterns were more “informative”. In relation to the same point, the difference between PC and SA1-like neurons in Fig. 3B is not obvious. This is another example where a statistical comparison of the two distributions is needed to support this suggestion; in the absence of any difference, then the statement should be removed.

We have described our statistical tests as follows:

“First, we found that spiking patterns of PC-like cortical cells were far more informative than were those of SA1-like cortical cells, as expected given the relative propensities of PC and SA1 nerve fibers to exhibit temporal patterning (Mackevicius et al., 2012; Weber et al., 2013) (1-sided Mann-Whitney U test comparing the best classification performance of PC-like and SA1-like neurons across resolutions, $U = 330$, $P < 0.001$; Figure 3A). Notably, both PC-like and SA1-like cells were more informative than were the rate-matched simulated Poisson models (Wilcoxon signed-rank test, PC-like: $U = 78$, $P < 0.001$; SA1-like: $z = 4.4$; $P < 0.0001$). Second, the informativeness of the responses of PC-like neurons always peaked at high temporal resolutions while not all SA1-like responses did, however these distributions were not significantly different (Kolmogorov-Smirnov test, $D=0.15$, $P= 0.65$; Figure 3B). Note that the high temporal resolutions of many SA1-like neurons may reflect the contribution of (non-dominant) PC or RA input or the maintained temporal reliability of exceptionally precise SA1 input (Supplemental Figure 8). “

Supplemental Fig. 6: I would not describe a sample of 7 PC afferents as a population. The term “population” must be removed from the title (B).

We have modified this to say “subpopulation”. We want to delineate classification schemes that rely on single cells vs groups of cells, and that is what we are using the term subpopulation to indicate.

Ln219: it would be better to refer to the “peripheral afferents” rather than the “nerves”.

We have reworded this to say, “Next, we compared the temporal coding in cortex to its peripheral counterpart”

Ln 279-81: This statement needs statistical support (higher level with both codes vs rate or timing alone).

We have included statistical support in the following:

“At the population level, combining rate and (warped) timing codes yielded better classification performance than did either code in isolation (permutation test comparing population classification using both rate and timing to rate alone, $P < 0.0001$, or to timing alone, $P < 0.0001$; Figure 5D, Supplemental Figure 12). This improvement could not be accounted for by the increase in the number of predictors (Supplemental Figure 11), indicating that spike timing carries information beyond that carried in the spike rates, even in larger populations of neurons (Figure 5D).”

Fig. 6A. The title refers to timing similarity while the x-axis label refers to timing correlation. Which term is correct?

We have updated the figure title to read “timing correlation”.

Methods. Were the individual textures presented in blocks of trials, or were the various textures presented in a pseudo random order? Please clarify.

We have clarified this with the following description:

“Textures were presented once each in pseudo-random order in each of 5 blocks and the order of presentation changed from block to block.”

Suppl Table 1. Why does this list of textures include many that were not used in any of the recordings. I suggest omitting these.

The table includes a complete list of the textures used in the cortical experiment presented. Some of these textures had not been used in our previous experiments (peripheral neurophysiology or dissimilarity

psychophysics), and that is why they do not have any 'x' marks in the following columns. But every texture listed was a part of the cortical analyses. We have tried to make this clearer in the legend:

"Column 1 includes all 59 textures presented during cortical neurophysiological recordings. Some of these textures had also been used in previous experiments: Column 2 shows the 24 textures that overlapped with those used in the peripheral experiments. Column 3 shows the 13 textures used in the psychophysical experiments. Column 4 shows the 10 textures that were presented at 4 speeds (60, 80, 100, 120 mm/s)."

Ln372-3. This observation (smooth vs textured) was first made by Depeault et al. (2008). Moreover, the results are complex: when roughness was systematically changed while keeping the material properties of the surfaces identical, a potential confound in this study, then surfaces were rated as moving slower for the roughest surface. It is likely that the smoothest surface tested here (chiffon) was ineffective in activating peripheral afferents. The text must be modified to include explicit recognition and discussion of the work by Depeault et al.

Thanks for pointing out this omission. We have added the following discussion of the relevant work by Depeault et al:

"Scanning speed-related signals in the nerve must thus somehow be discounted in the computation of texture. This process is not trivial, as evidenced by the fact that the converse is not true: Tactile speed perception is highly dependent on texture (Delhaye et al., 2019; Dépeault et al., 2008)."

Suppl Fig 11: The title over each plot is incorrect (y is plotted as a function of x, and not vice-versa as here). Please change the order for each title. This same issue also arises for Fig. 6A. Further, what is the difference between perceptual roughness and perceptual dissimilarity? Finally, the legend should specify numbers of subjects and texture pairs contributing to the plots.

Thank you for pointing this out; we have corrected the titles of the figures in Sup Fig 11 and Fig 6A.

We outline perceptual roughness and dissimilarity in the text and in the methods. In brief, roughness is a perceptual dimension of texture, but it is not the only one. In two separate experiments, we asked subjects to rate (1) the roughness of individual textures and (2) the dissimilarity between pairs of textures. Two textures could be equally rough but feel very different from one another along another dimension, say for example by differing in compliance.

We have added the details to the legend:

Supplemental Figure 13 (formerly 11): ". **A**] The absolute difference in mean firing rate across the full population of 141 cortical cells vs. the absolute difference in roughness ratings (rated by 6 human subjects) for the same texture pairs (78 unique pairs). Differences in firing rate predict differences in perceived roughness. **B**] Differences in firing rate only account for 20% of the variance in the dissimilarity ratings (rated by 10 human subjects)."

Figure 6A: "**C**] Accuracy of the optimal model (average n=12 population response of SA1 rate and PC timing) in predicting perceived dissimilarity. Each circle represents one texture pair (78 unique pairs, rated by 10 human subjects)."

REVIEWER COMMENTS

Reviewer #1 (Remarks to the Author):

I am satisfied with the changes made to address my comments and those of the other reviewers.

Supp Fig 13 has "perceptual" mis-spelled in several places.

Reviewer #2 (Remarks to the Author):

Authors had solved all our concerns and those of the other two reviewers.

We understand the authors tried their best on including all the data from previously neglected populations (rapid-adapting afferent dominant) in the revised manuscript.

We have no further suggestions to add and recommend this manuscript for publication.

Reviewer #3 (Remarks to the Author):

This is the revised version of a paper that I recently reviewed. The authors have made some changes to address my concerns, but my first major criticism was not adequately addressed.

Major

Need for a more balanced view to the cortical encoding of tactile texture. The authors claim that they modified Figures 3 and Suppl Fig 8 to show the "other" cells rather than all cells. A comparison of the figures shows that this is not true (apart from one exception, Suppl Fig. 9C): the plots are identical to the original, despite the change in label. As I explained before, this is misleading as the plots are dominated by SA- and PC-like cells, so that the relative importance of the "other" cells, many of which undoubtedly received input from RA receptors (rapidly adapting), is diminished. I suspect that the results would have been similar to those shown in Fig. 4A which plots the results of all cells as a function of the cytoarchitectonic area. Here we can see that the area 3b cells (SA and "other"; no PC) perform better than either area 1 (includes PC, SA and "other") or area 2 (2 PC, SA and "other"). The critical importance of the "other" cells is clearly illustrated in Suppl Fig. 9C which shows better classification performance for these cells than for either SA or PC-like cortical cells. Unfortunately, this information is buried in the supplemental data and not even mentioned, let alone discussed, in the text. The authors must modify Fig. 3 A and B, showing the "other" cells rather than all cells. A fifth panel, E, should be added to show the "other" cells, complementing what is shown in C (SA-like) and D (PC-like). Suppl Fig. 8 also needs to be revised following the same approach ("other" not all where appropriate). These results must then be described in the text and discussed.

In passing, the authors argue that they cannot categorize the "other" neurons, but they could easily run the same regression analysis as for SA- and PC-like neurons, using their population of RA afferent recordings, only retaining those with the highest correlation.

Finally, I would add that the title of the paper is misleading: this only refers to timing but the authors conclude that both rate and timing information contribute to cortical encoding of roughness.

Specific points

1. My original point 5 was not adequately addressed in the text. There is no need to invoke downstream transformations in secondary somatosensory cortex, as the proposed transformation is already present in area 3b and 1 where roughness is encoded independent of speed: the authors now cite Bourgeon et al (2016) but not in the context of rate coding in primary somatosensory cortex independent of scanning speed.

Minor

The text describing the human psychophysical testing in the Results remains confusing and

incomplete (see also reviewer 2) as there is no reference to the fact that 2 series of experiments were performed (roughness rating, roughness dissimilarity) and used to evaluate the putative neuronal codes (pg7, paragraphs 4-5).

Reviewer #1 (Remarks to the Author):

I am satisfied with the changes made to address my comments and those of the other reviewers. Supp Fig 13 has "perceptual" mis-spelled in several places.

Good catch, thank you!

Reviewer #2 (Remarks to the Author):

Authors had solved all our concerns and those of the other two reviewers.

We understand the authors tried their best on including all the data from previously neglected populations (rapidly adapting afferent dominant) in the revised manuscript.

We have no further suggestions to add and recommend this manuscript for publication.

Thank you!

Reviewer #3 (Remarks to the Author):

This is the revised version of a paper that I recently reviewed. The authors have made some changes to address my concerns, but my first major criticism was not adequately addressed.

Major

Need for a more balanced view to the cortical encoding of tactile texture. The authors claim that they modified Figures 3 and Suppl Fig 8 to show the "other" cells rather than all cells. A comparison of the figures shows that this is not true (apart from one exception, Suppl Fig. 9C): the plots are identical to the original, despite the change in label.

We did change all cells to other cells in the first resubmission. The reason the plots look similar is that the difference between "all" cells and "other" cells is the removal of the SA1-like and PC-like neurons, which are at the extremes of the continuum. When we remove these extreme cells, the mean curve looks similar to that when we keep them in.

As I explained before, this is misleading as the plots are dominated by SA- and PC-like cells, so that the relative importance of the "other" cells, many of which undoubtedly received input from RA receptors (rapidly adapting), is diminished. I suspect that the results would have been similar to those shown in Fig. 4A which plots the results of all cells as a function of the cytoarchitectonic area. Here we can see that the area 3b cells (SA and "other"; no PC) perform better than either area 1 (includes PC, SA and "other") or area 2 (2 PC, SA and "other"). The critical importance of the "other" cells is clearly illustrated in Suppl Fig. 9C which shows better classification performance for these cells than for either SA or PC-like cortical cells. Unfortunately, this information is buried in the supplemental data and not even mentioned, let alone discussed, in the text. The authors must modify Fig. 3 A and B, showing the "other" cells rather than all cells. A fifth panel, E, should be added to show the "other" cells, complementing what is shown in C (SA-like) and D (PC-like). Suppl Fig. 8 also needs to be revised following the same approach ("other" not all where appropriate). These results must then be described in the text and discussed.

In passing, the authors argue that they cannot categorize the "other" neurons, but they could easily run the same regression analysis as for SA- and PC-like neurons, using their population of RA afferent recordings, only retaining those with the highest correlation.

In this revision, we have replaced "other" cells with "RA-like" cells in Figure 3 and Supplemental Figures 8, 9, 13, and 14. This population of neurons is slightly more informative and higher in temporal resolution than was the "other" population. Otherwise, the results are similar.

Finally, I would add that the title of the paper is misleading: this only refers to timing but the authors conclude that both rate and timing information contribute to cortical encoding of roughness.

The informativeness of rate about texture has been previously demonstrated. The title highlights the novel aspect of the present study.

Specific points

1. My original point 5 was not adequately addressed in the text. There is no need to invoke downstream transformations in secondary somatosensory cortex, as the proposed transformation is already present in area 3b and 1 where roughness is encoded independent of speed: the authors now cite Bourgeon et al (2016) but not in the context of rate coding in primary somatosensory cortex independent of scanning speed.

We found that the rate-based texture signal in somatosensory cortex is **not** independent of speed when tested with a wide range of textures (Lieber and Bensmaia, Cerebral Cortex, 2020), though more so than its counterpart in the nerves. The discrepancy is likely due to the fact that Bourgeon et al. used embossed dot patterns, which are strongly encoded in spatial patterns of activation. The representation of such patterns is far less reliant on a temporal code than is that of finer textures and thus less susceptible to variations in speed.

Minor

The text describing the human psychophysical testing in the Results remains confusing and incomplete (see also reviewer 2) as there is no reference to the fact that 2 series of experiments were performed (roughness rating, roughness dissimilarity) and used to evaluate the putative neuronal codes (pg7, paragraphs 4-5).

Thanks for pointing this out. We have now specified that there are two tasks:

“To test this, we asked human subjects to perform two psychophysical tasks: Roughness scaling and dissimilarity scaling. In the roughness scaling task, each texture was presented individually and the subject provided a rating of its roughness. In the dissimilarity scaling task, a subset of thirteen textures was presented in pairs and the subject rated the dissimilarity of each pair (Lieber and Bensmaia, 2019) (Supplemental Table 1). While roughness ratings gauge how textures fit along a single sensory continuum, dissimilarity ratings take into account every way in which textures might differ.”

REVIEWER COMMENTS

Reviewer #3 (Remarks to the Author):

This is the 2nd revision of a paper that I reviewed.

The authors have partly addressed my criticisms, and now include a sample of 82 RA-like (rapidly adapting) neurons, chosen using the same criteria applied to the PC- (Pacian, n=12) and SA-like (slowly adapting, n=25) neurons. Note that the large RA-like population of neurons (~70% of the sample) was ignored in the original submission and still is in this version, apart from appearing in 5 figures (4 of 5 buried in the supplemental data). The RA-like neurons are still ignored in the results and discussion which concentrate on a comparison between PC- and SA-like neurons. I found only two references to RA-like neurons in the text (excluding figure legends). In short, the authors continue to ignore the contribution of RA-like cortical neurons to the encoding of tactile texture.

Specific.

I note that the authors did not, as I requested, add a panel to Fig. 3 to plot the population classification for RA-like cells, as done for SA- and PC-like cells (Fig 3CD). This must be addressed. I expect that this will show the similarity between the PC-like and RA-like cells in relation to the main point of this paper – rate and timing coding of tactile texture.

Thank you for these comments. The reviewer was absolutely justified in insisting that we include a more detailed analysis of the responses that receive strong input from RA fibers. Upon reflection, we realized that there was no good justification for excluding these or downplaying their importance. In the newest revision, we have remedied this failing by analyzing RA-like neurons wherever submodality input is discussed, and the paper is better for it. Furthermore, in the process of re-running some of the analyses, we caught an error in the indexing of RA-like neurons: We had accidentally set the threshold too low in the previous iteration. The more exclusive criterion (which matches the criterion used to identify SA1-like and PC-like neurons) yields similar conclusions, though the responses of the 12 RA-like neurons are more temporally patterned than were those based on the more inclusive criterion. The picture that emerges is that temporal patterning in RA-like neurons is intermediate between that of PC neurons and SA1-like neurons, which is illustrated in Figure 3. Interestingly, though, timing in RA-like neurons is not predictive of perception. We are grateful for the reviewer's diligence and perseverance, as we feel the revised manuscript is now balanced and thus improved. We hope you will too.

This is the 2nd revision of a paper that I reviewed.

The authors have partly addressed my criticisms, and now include a sample of 82 RA-like (rapidly adapting) neurons, chosen using the same criteria applied to the PC- (Pacinian, n=12) and SA-like (slowly adapting, n=25) neurons. Note that the large RA-like population of neurons (~70% of the sample) was ignored in the original submission and still is in this version, apart from appearing in 5 figures (4 of 5 buried in the supplemental data). The RA-like neurons are still ignored in the results and discussion which concentrate on a comparison between PC- and SA-like neurons. I found only two references to RA-like neurons in the text (excluding figure legends). In short, the authors continue to ignore the contribution of RA-like cortical neurons to the encoding of tactile texture.

Specific.

I note that the authors did not, as I requested, add a panel to Fig. 3 to plot the population classification for RA-like cells, as done for SA- and PC-like cells (Fig 3CD). This must be addressed. I expect that this will show the similarity between the PC-like and RA-like cells in relation to the main point of this paper – rate and timing coding of tactile texture.

REVIEWERS' COMMENTS

Reviewer #3 (Remarks to the Author):

The authors have finally addressed my criticisms. Thank you, and I agree that the manuscript is much improved and balanced.

I have only one question: is the legend for Supp. Figure 8B correct? I would have expected higher values for PC (vs RA) afferents.

Response to Reviewer

Reviewer #3 (Remarks to the Author):

The authors have finally addressed my criticisms. Thank you, and I agree that the manuscript is much improved and balanced.

I have only one question: is the legend for Supp. Figure 8B correct? I would have expected higher values for PC (vs RA) afferents.

Thank you again for your helpful feedback! Yes, the legend for Figure 8B is correct. The rapidly-adapting afferents are very reliable across repeats, but this observed temporal precision is not maintained in cortex to the same degree as that of the PC afferent inputs to cortex.